# Assessing the response of micro-eukaryotic diversity to the Great Acceleration using lake sedimentary DNA

François Keck [1,2], Laurent Millet[3], Didier Debroas[4], David Etienne[5,2], Didier Galop [6,7], Damien Rius[3] & Isabelle Domaizon [1,2✉]

Long-term time series have provided evidence that anthropogenic pressures can threaten lakes. Yet it remains unclear how and the extent to which lake biodiversity has changed during the Anthropocene, in particular for microbes. Here, we used DNA preserved in sediments to compare modern micro-eukaryotic communities with those from the end of the 19th century, i.e., before acceleration of the human imprint on ecosystems. Our results obtained for 48 lakes indicate drastic changes in the composition of microbial communities, coupled with a homogenization of their diversity between lakes. Remote high elevation lakes were globally less impacted than lowland lakes affected by local human activity. All functional groups (micro-algae, parasites, saprotrophs and consumers) underwent significant changes in diversity. However, we show that the effects of anthropogenic changes have benefited in particular phototrophic and mixotrophic species, which is consistent with the hypothesis of a global increase of primary productivity in lakes.

[1] INRAE, Université Savoie Mont Blanc, CARRTEL, 74200 Thonon-les-Bains, France. [2] Pole R&D ECLA, CARRTEL, 74200 Thonon-les-Bains, France. [3] CNRS, Chrono Environnement, 25000 Besançon, France. [4] Université Clermont Auvergne, CNRS, Laboratoire Microorganismes: Genome et Environnement, 63000 Clermont-Ferrand, France. [5] Université Savoie Mont Blanc, INRAE, CARRTEL, 73370 Le Bourget du Lac, France. [6] GEODE UMR 5602 CNRS, Université de Toulouse, 31058 Toulouse, France. [7] Labex DRIIHM, OHM Pyrénées, CNRS/INEE, Toulouse, France. ✉email: isabelle.domaizon@inrae.fr

Lakes are highly exposed to human- and climate-induced pressures. Their ecological functioning, the biodiversity they host and the ecosystem services they provide are threatened by water pollution, resource exploitation, degradation of habitats, and climate change effects among other stressors[1–3]. Anthropogenic activities have impacted lakes for millennia[4,5], but pressures have largely increased in magnitude since the mid-twentieth century, i.e., The Great Acceleration[6,7]. The second part of the twentieth century is unambiguously a time of major and multiple threats that might have disproportionately affected lake biodiversity and associated ecological functions (e.g., nutrient recycling, efficiency of trophic transfer, and quality of fish production)[3,8]. Paleolimnological studies have provided evidence supporting the occurrence of states of change in lake ecosystems, for instance the accelerated hypoxia in European lakes after 1900[9] or the increase in autochthonous primary production in lakes (e.g., Lake Superior[10] and see review by Smol[11]). However, much less is known about the long-term changes of biodiversity. If several anthropogenic stressors have been shown to influence the diversity of amphibians, fish, macro-invertebrates, or water birds[3], there is a paucity of information regarding microbial diversity responses.

Among microorganisms, micro-eukaryotes are a large and diversified group both in terms of taxonomy and functional roles; they act as primary producers, consumers, parasites, and saprotrophs in planktonic assemblages. Being recognized as relevant ecological indicators[12], they are good candidates for investigating the impact of local and global changes on lacustrine biodiversity. Until recently, only few micro-eukaryotic groups able to produce morphological fossils (e.g., diatoms, chrysophytes) and specific pigments, were considered in paleolimnological studies. With the development of metabarcoding and high-throughput sequencing (HTS) applied to DNA preserved in sediments, it is now possible to reconstruct past microbial assemblages and infer historic ecological networks[13]. Despite technical difficulties related to the recovery of short and potentially degraded DNA fragments[14,15], recent works have demonstrated the relevance of sedimentary DNA to investigate temporal changes in micro-eukaryote assemblages and their responses to anthropogenic and climatic pressures[16–21]. Although these applications were limited to only a few lakes, they paved the way for exciting future applications in paleolimnology. The coupling between paleolimnological approaches to study the magnitude and timing of change for lake ecosystems, and the potential of sedimentary DNA to reveal the composition of past biological assemblages, offers new opportunities to track historical conditions of lacustrine biodiversity. Due to a lack of long-term monitoring data, we must establish pre-impact conditions (between 1750 and 1900[22]) as a reference to assess how microbial communities have changed with accelerated human pressures.

Here, we used DNA records preserved in sediment to explore pre-impact (nineteenth century) lake conditions chosen before the "Great Acceleration", a turning point in the Anthropocene from which the human imprint on limnic ecosystems became observable[7]. We analyzed sediment cores from 48 temperate lakes to compare modern micro-eukaryotic communities with assemblages from the nineteenth century, using a so-called top-bottom paleolimnological approach[23], whereby DNA preserved in recent sediment deposits (i.e., top) representing modern biological assemblages is compared to a sediment sample representing reference conditions from the nineteenth century (i.e., bottom). The 48 study sites (located in France; Supplementary Fig. 1 and Supplementary Table 1) covered different lake typologies and a wide elevation gradient, from lowland to high-elevation lakes. Micro-eukaryotic assemblages were reconstructed from sedimentary DNA using HTS of 18S rDNA. In order to investigate the magnitude of compositional change, we compared microbial diversity from recent and pre-impact periods. For each lake, different ecological metrics were considered, taking into account the composition in operational taxonomic units (OTUs) for the whole micro-eukaryotic assemblage, but also for each taxonomic and functional group (photosynthetic, mixotrophs, parasites, saprotrophs, and consumers). We tested whether the changes in community composition modified the similarity of microbial assemblages among lakes (i.e., spatial turnover), in particular, to explore the potential biotic homogenization among lakes due to global pressures[24]. For each taxonomic and functional group, we performed differential abundance analyses between recent (i.e., the top sediment sample) and past (i.e., the bottom sediment samples) to elucidate which taxonomic groups contributed the most to the community changes.

Together, our results provide an unprecedented picture of the changes in lacustrine micro-eukaryotic diversity since the nineteenth century. DNA preserved in lake sediment is an essential tool to complement traditional proxies and study biological diversity dynamics and ecosystem trajectories in a context of environmental change.

## Results

**Overview of sediment records**. Based on chronological information obtained for each core (see "Methods" section and Supplementary Table 2), two sediment layers, i.e., a recent sediment deposit and a sediment deposit from the nineteenth century, were subsampled and preserved for downstream analyses. Strict protocols have been applied to ensure the robustness of DNA results (Supplementary Methods). The quantity and quality of DNA extracted from sediments were found to meet the requirements for the reconstruction of micro-eukaryotic diversity for 48 lakes out of the 53 lakes initially sampled (Supplementary Table 1).

Sedimentary proxies for organic (i.e., total organic carbon) and biogenic (i.e., pigments and DNA) compounds were found in larger quantities in recent than in past strata (Fig. 1; all paired Wilcoxon test $p$-values < 0.001). This suggests a lower biological production in the past, as long as the diagenetic processes occurring during burying in sediments had a limited effect on the loss of organic material.

After all the filtering and processing steps applied for metabarcoding raw data (see "Methods" section and Supplementary Fig. 2), 2,321,458 DNA reads were retained and clustered into 2451 OTUs (95% identity). Most of these OTUs could be classified into a defined taxonomic group of the Adl et al. classification[25]: 95.4% into 14 groups at the 2nd rank and 74.8% into 29 groups at the 3rd rank. Most of them (73.9%) could also be assigned to one of the five trophic groups (consumers, saprotrophs, parasites, photosynthetics, and mixotrophs). Alveolata, Rhizaria, Stramenopiles, and Opisthokonta were dominant in terms of richness (i.e., number of OTUs). The 14 taxonomic groups were retrieved both in the recent and past strata, and the richness distribution within these groups was globally stable in top and bottom strata (Supplementary Fig. 3). Many OTUs (1961; 80%) were detected both in the recent and past strata. However, some OTUs (80; 3%) were found only in the past samples and others (410; 17%) only in the recent samples (Supplementary Fig. 4).

**Temporal and spatial beta-diversity**. Multivariate analysis of beta-diversity was conducted using both Jaccard (presence/absence of OTUs) and Bray–Curtis (relative abundances of OTUs) indices. Results were highly similar; only the detailed results corresponding to Bray–Curtis dissimilarities are therefore

reported here (results corresponding to Jaccard dissimilarities are presented in Supplementary Fig. 5).

Bray–Curtis dissimilarities between past and recent samples ranged from 0.10 to 0.96 with a mean of 0.74 (Fig. 2b). The multivariate analysis of variance (PERMANOVA) conducted on Bray–Curtis dissimilarities indicated a difference in OTU composition of micro-eukaryotic communities between the two groups (i.e., recent and past; $p$-value < 0.001, see Supplementary

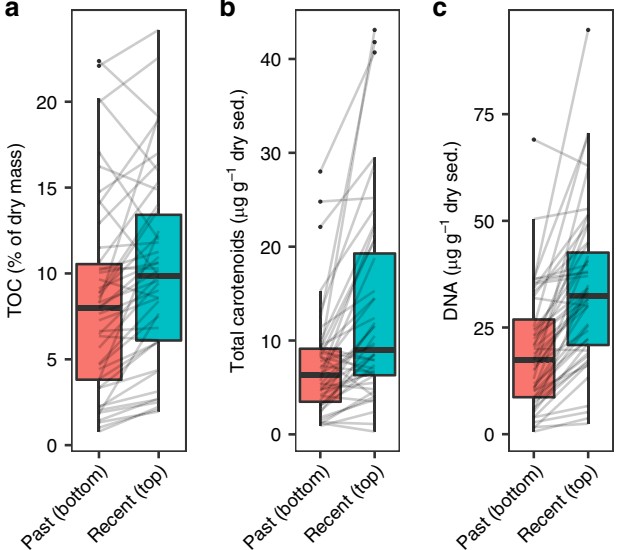

**Fig. 1 Differences between the recent (top) and past (bottom) sediment strata. a** Proportion of total organic carbon (TOC) in dry sediments ($n = 46$ lakes), **b** concentrations of total carotenoid pigments expressed in µg per g of dry sediment ($n = 44$ lakes) and **c** concentrations of DNA expressed in µg per g of dry sediment in the recent (top) and past (bottom) strata ($n = 48$ lakes) of the sediment cores. Boxplots depict medians, first and third quartiles, and full ranges (bounded at 1.5 × interquartile range). Gray lines connect values from the same lake.

Table 3). The shift in community composition is clearly displayed by nonmetric multidimensional scaling analysis (NMDS; Fig. 2a), where the recent samples are distinctly separated from the past samples for most of the lakes.

The NMDS plot (Fig. 2a) also suggests a strong reduction of spatial beta-diversity (i.e., among lakes) denoted by a reduction in sample dispersion and confidence ellipse size between recent and past samples. This result is supported by the paired test for homogeneity of multivariate dispersions. The average distance of samples to their geometric median was 0.606 for the recent samples and 0.678 for the past samples (Wilcoxon test $p$-value < 0.001, see Fig. 2c).

Considering the elevation of lakes, the regression tree model identified one significant split in the data at ~1400 m, indicating that magnitudes of change for micro-eukaryotic communities were higher in low/moderate elevation lakes than those estimated for high-elevation lakes (>1400 m). On average, the Bray–Curtis dissimilarity index between past and recent was 0.85 for lakes below 1400 m and 0.53 above (Fig. 3). The two groups of lakes also exhibited very different variances, with much more variable Bray–Curtis index values for high-elevation lakes (Fig. 3, $F$-test for equality of variance: variance ratio = 0.11; $p$-value < 0.001).

To circumvent potential limits associated with the analysis of OTUs (Supplementary Methods), we replicated the analyses at a coarser taxonomic grain, considering the 14 taxonomic groups (2nd rank of Adl et al. classification[25]). Similar patterns were obtained with these aggregated data (Supplementary Fig. 6), leading to the same conclusions regarding the marked temporal changes for most of lakes, in particular below 1400 m of elevation, and the reduction in beta-diversity.

**Differential abundance analysis.** The differential abundance analysis (DESeq2) found a significant increase of species capable of photosynthesis in the recent strata (Fig. 4; Supplementary Table 4) with an increase of obligate photoautotrophs, i.e., photosynthetics ($p$-value = 0.022) and mixotrophs ($p$-value < 0.001). In addition, we observed a significant reduction of heterotrophic

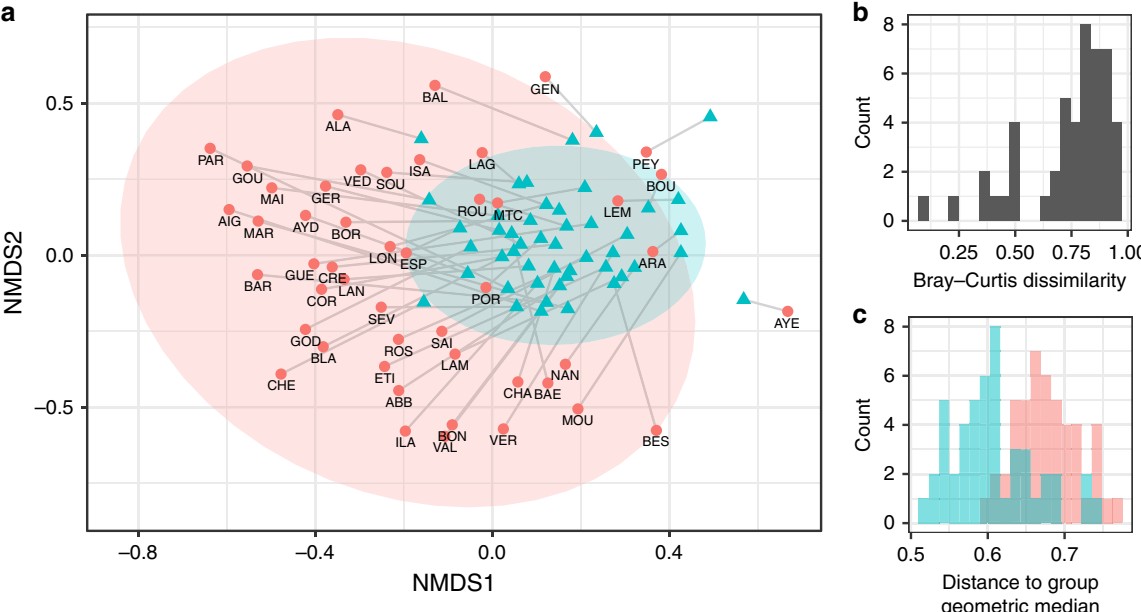

**Fig. 2 Multivariate analysis of micro-eukaryotic communities based on OTUs matrix. a** NMDS of community compositions of the recent (blue triangles) and past (red circles) samples with 95% confidence ellipses represented for each group. Gray lines connect recent and past samples from the same lake. Stress = 0.24. **b** Distribution of Bray–Curtis index values computed for each lake between recent and past samples. **c** Distribution of the distances between samples and group geometric median for recent (blue) and past (red) samples.

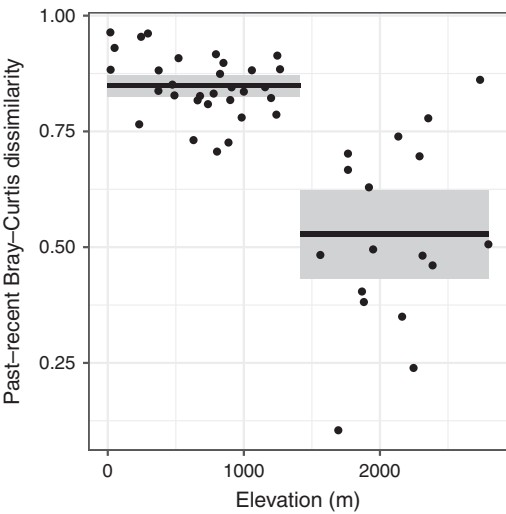

**Fig. 3 Relationship between lake elevation and temporal turnover.** Fitted regression tree model (*n* = 48 lakes) is represented by black lines (mean values). Gray shading represents the 95% confidence intervals around means.

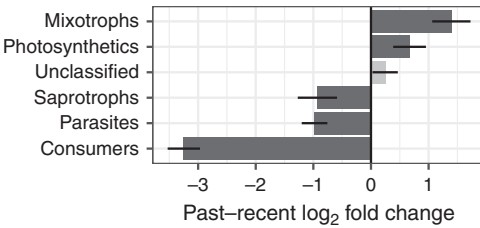

**Fig. 4 Magnitude of change in trophic groups between the past and recent strata.** Magnitude of change is expressed in log$_2$ fold change, as estimated by the DESeq2 analysis[75] (*n* = 48 lakes). Dark bars represent groups for which the change was found significant (two-sided Wald test corrected with the Benjamini & Hochberg method *p*-value < 0.05). The change was found significant for mixotrophs (*p*-value < 0.001), photosynthetics (*p*-value = 0.022), saprotrophs (*p*-value = 0.009), parasites (*p*-value < 0.001), and consumers (*p*-value < 0.001). The change was not found significant for the unclassified group (*p*-value = 0.263). Horizontal lines show the standard error.

species mostly represented by consumers (bacterivores, algivores, and omnivores; *p*-value < 0.001) but also by saprotrophs (*p*-value = 0.009) and parasites (*p*-value < 0.001). To identify which taxonomic groups were involved in these temporal changes, the DESeq2 analysis was also conducted on reads aggregated by taxonomic groups. We found 14 heterotrophic groups for which the number of reads significantly decreased in the recent period compared to the past (Supplementary Fig. 7 and Supplementary Table 5), four of which belong to Cercozoans (Glissomonadida, Paracercomonadida, Thecofilosea, and Cercomonadida). These groups are mostly known as phagotrophic predators on bacteria and are both constituents of pelagic and benthic food webs. For few heterotrophic groups, as for instance Peronosporomycetes (known for their role as saprotrophs or plant parasites), the analysis showed an increase of the number of reads in recent period. However, as mentioned before, this positive trend was observed mainly for photosynthetic or mixotrophic groups, in particular for Prymnesiophyceae, Dinophyceae, and Chlorophyceae whose relative proportions increased significantly in the recent period.

## Discussion

We used DNA archived in sediment to analyze the changes in lake microbial diversity during the last ~150 years. This period includes the Great Acceleration, which saw human pressures on the Earth System increase substantially, to the point of markedly altering climate, geochemical cycles, and biodiversity. Significant modifications in environmental conditions induced by human activities during this period are presumably the main driver of changes observed globally in the community composition of mammals, birds, fish, invertebrates and plants[26]. Our results are in line with these findings and indicate drastic changes in the composition and structure of micro-eukaryotic communities. Though microbes represent the largest source of biodiversity and ecological functions, they are often omitted from debates concerning global biodiversity policy[27]. Due to their rich diversity, rapid generation time and ubiquitous distribution, microbial communities are still often considered as functionally redundant and omnipresent; consequently microbial diversity is generally absent in the ongoing debates about global biodiversity modification[28]. There is however growing evidence showing that microbial communities can be sensitive to disturbances, which in turn influence ecosystem recovery[29]. The application of DNA-based methods in paleolimnology can help to gain knowledge on the long-term dynamics of aquatic microbial communities. Here, we demonstrated that DNA preserved in sediment not only allows us to expand the range of organisms detectable from natural archives, but also reveals multiple dimensions of biodiversity from OTUs to functional groups. This meets the need we have to understand biological responses at high levels of organization (i.e., communities, food webs, and ecosystem)[30].

We were able to show that for the majority of our study lakes, modern micro-eukaryotic communities differ from what we considered to be their "reference state" and that the direction of microbial turnover is globally consistent across lakes. The magnitude of temporal turnover between pre-impact and modern communities is particularly high. Comparable changes in magnitude have been reported in two peri-alpine lakes where high Bray–Curtis dissimilarity values (~0.8) were found over the last century[17,18]. These observations highlight the need to (i) collect more data to analyze the response of micro-eukaryotes to a wide range of environmental stressors and (ii) investigate more precisely the relationship between micro-eukaryotic assemblages and lake ecological functions and processes.

Though the top-bottom approach is limited to two periods of time, we demonstrate the value paleogenetic tools can add to assess how aquatic biodiversity has changed from its pre industrial state. We showed that the magnitude of temporal turnover varies with lake elevation, the difference between past and modern community composition being much more pronounced in low elevations. High-elevation locations are assumed to be less exposed to local anthropogenic pressures[31,32]. For lakes in particular, the low human density and reduced agricultural activity at high elevations (Supplementary Fig. 8) strongly limit the risk for direct anthropogenic nutrient enrichment[33]. Thus, we provide support that the temporal turnover observed is primarily driven by local human activities rather than long-term ecological drift. It is worth noting that this result does not contradict the idea that high-elevation lakes might be particularly sensitive to climate-related changes[34]. We indeed observed high turnover for a few elevation lakes (for instance two high-elevation lakes in the Massif Mont Blanc). As previously reported, moderately nutrient-enriched Alpine lakes are likely to experience increases in their productivity with climate warming (e.g., Sadro et al.[35]). Temporal changes of beta-diversity observed at the community level are the result of the loss of some taxa, the emergence of others, and changes in relative frequencies. Differential abundance analysis

identified several eukaryotic groups that markedly increased or decreased during the last century. These changes in abundance are likely driven by modifications in environmental conditions that select species with different ecological requirements. Previous studies have used sedimentary DNA to show temporal shifts in taxonomic and functional diversity of micro-eukaryotes in direct relation to changes in environmental conditions whether it be oxygen depletion in the Arabian Sea in relation to monsoon cycles[36] or human-induced eutrophication and heavy metal pollution in Lake Chao in China[37]. Eutrophication has also been identified as the major driver of micro-eukaryotic diversity in two European deep lakes over the last century[18], while statistical models showed that recent warming also represented a significant driver of biodiversity change after the mid-80s. In addition to abiotic conditions, temporal turnover of communities could also be influenced by the abundance of predators (zooplankton) via top-bottom control. Biotic interactions also occur within the total micro-eukaryotic community itself, which is organized into a complex food web, with several groups playing distinct trophic functions. However, nutrient enrichment and climatic warming undoubtedly represent two major drivers in lakes functioning[38], with impacts on the magnitude of phytoplankton blooms[39] as reported for Cyanobacteria in particular[40]. Our results are in line with the hypothesis of an overall increase of photosynthetic species over the last century, and showed that not only cyanobacteria but also eukaryotic pigmented groups respond to these stressors. Respondent groups include both obligate oxygenic photoautotrophs (in particular Chlorophyceae and Prymnesiophyceae) and mixotrophs (in particular Dinophyceae). Our combined DNA and pigment-based results obtained for 48 lakes are in line with several paleolimnological studies that have reported an increase in primary production throughout the Holocene[10,11]. The ratio and rates of autotrophic and heterotrophic production constitute the basis of the metabolic balance of lake ecosystems in the form of primary productivity and respiration. We therefore hypothesize that the metabolic balance of lakes has been modified towards an increase of phototrophic processes in recent periods, with potential impact on carbon fixation in lakes[41]. If confirmed, this would mean that anthropogenically driven changes in these biological assemblages might have affected the carbon cycle in lakes, with a potential contribution to the increase of lake C burial observed from the late nineteenth century[42].

The distinct taxonomic groups forming the primary producer community respond differently to environmental conditions according to their physiological capacities. Standardized phytoplankton-based indices used to evaluate lake trophic status generally use Chlorophyceae as a typical algal groups for eutrophic conditions (with Cyanobacteria, and Euglenophyceae, e.g., Laplace-Treyture and Feret[43]). The success of Chlorophyceae could therefore be explained by the global increase of nutrients in lakes over the last century[9], as revealed by both limnological[44,45] and paleolimnological records[46–48]. For example, in the paleolimnological study from Capo et al.[18], a strong response of Chlorophyceae to [P] concentration was recorded in two alpine lakes, with an increase of their relative proportion since the 1960s. Conversely, the mixotrophic group Dinophyceae, which also increased proportionally in the recent period, is generally known to prevail in more oligotrophic lakes. Dinophyceae community structure is driven by a combination of factors including temperature, light, nutrient inputs and climatic fluctuations[49]. Mixotrophy is a common trophic strategy among unicellular eukaryotes. Although the harsh growth conditions found in high mountain lakes seem in particular to favor mixotrophs[50], there is currently little evidence for how mixotrophy varies across environmental gradients[51].

In addition to the changes in composition, we report that the average similarity among lake communities significantly increased in the recent period, suggesting a global homogenization trend of the micro-eukaryotic communities during the last 150 years. Biotic homogenization has been reported in many biological groups and ecosystems as a result of anthropogenic activities and urbanization[52–55], and freshwater ecosystems are particularly affected by this phenomenon[54,56]. There are limited data available regarding biotic homogenization in microbial communities, however Monchamp et al.[57] reported evidence of phylogenetic homogenization in lake cyanobacterial communities across ten European peri-alpine lakes. Biotic homogenization can be the consequence of a reduction in regional environmental heterogeneity selecting for a reduced pool of species and traits[52,58]. For example, homogenization of cyanobacterial communities in peri-alpine lakes has been linked to external forcings like eutrophication and climate change, which have led to a global homogenization of environmental conditions over the last 100 years[57]. In addition, biotic homogenization that depends on the ability of non-native species to disperse can be facilitated by modern transportation and increased connectivity between sites. The colonization of new habitats by microbial species has been influenced by these human activities both in terrestrial and aquatic systems[59,60].

A number of gaps exist in our understanding of microbial diversity patterns and their long-term dynamics. This is true in particular for heterotrophic protists, which are often missing in lake biomonitoring and aquatic biogeochemical models. Here we showed the possibility to track the past occurrence and dynamics of various heterotrophic groups, which are constituents of both pelagic and benthic food webs. This allows providing unique information in terms of lake functional ecology. For instance, our data suggest a decrease of taxa living on oxygenated surface sediments (e.g., Cercozoa) in recent periods, which tends to confirm that benthic food webs were affected by the general expansion of hypoxia in lakes during the last centuries[61].

The application of DNA-based methods in paleolimnology can certainly help to fill several gaps, in particular for gaining knowledge on assessing diversity, resistance, and resilience of microbial communities to environmental forcings and the links between microbial diversity and the ecological state of lakes. Combining traditional sedimentological and geochemical proxies with DNA data should be promoted to take full advantage of the complementarity of these indicators, whilst keeping in mind the potential strengths, weaknesses, and sensitivities of the different proxies[62].

## Methods

**Study sites and coring**. In each studied lake, a sediment core was retrieved from the deepest part of the lake basin (from 6 to 309 m below the water surface, see Supplementary Table 1) using a UWITEC gravity corer. In total 53 lakes were sampled; however DNA analyses were only possible on 48 lakes due to limited preservation of DNA in five sediment cores. In all cases, sediment cores were stored in a cold room (4 °C) in the darkness and air-protected by a double layer of plastic wrap and by a sealed plastic sheath.

**Dating and subsampling of sediment cores**. Two sediment layers were sampled at the top and the bottom of each lake sediment record to document the two targeted periods, i.e., the modern times and the nineteenth century. The stratigraphic positions along cores of these two samples were determined using a combination of several approaches (X-ray fluorescence, radiocarbon and radionuclides $^{210}$Pb and $^{137}$Cs) depending on lakes (Supplementary Table 2). The technical details about the dating method are given in (Supplementary Methods). The thickness of sediment samples was individually adjusted so that each sample covered at least 10–15 years. This sampling strategy was applied to provide an integrated picture of lake biodiversity over a significant duration, to smooth inter-annual variability and then to ensure a more robust comparison between lakes, as well as between top and bottom samples within each lake.

Subsampling consisted of three individual samples for organic matter, pigment and DNA analysis, respectively. To avoid contamination by modern DNA, subsampling operations were performed in a controlled environment and only the center of each core was used for molecular analysis after a careful cleaning of core surface (Supplementary Methods). Organic carbon contents of sediment samples were analyzed using a vario MAX CNS analyzer (Elementar) and photosynthetic pigments by spectrophotometry. In total, carotenoids were measured for 44 lakes and organic carbon for 46 lakes. Technical details are given in Supplementary Methods.

**Molecular analyses**. We applied strict laboratory protocols to ensure the validity of our molecular data (Supplementary Methods). For each sediment layer, two DNA extractions were performed on 0.5 g of wet sediment using the NucleoSpin® soil kit, according to the manufacturer instructions (Macherey-Nagel, Düren, Germany). The V7 region of the 18S rRNA gene was PCR amplified on a 260-bp-long fragment, from c.a. 25 ng of environmental DNA extracted from each sample and using the general eukaryotic primers 960 F (5′-GGCTTAATTTGACT-CAACRCG-3′)[63] and NSR1438 (5′-GGGCATCACAGACCTGTTAT-3′)[64]. The choice of the barcode region has been validated as reported by Capo et al.[17,18]. Details about the preparation of purified amplicons are given in Supplementary Methods.

The purified amplicons for each sample were pooled at equimolar concentrations and sent to GeT-PlaGe (Plateforme Génomique 31326 CASTANET-TOLOSAN Cedex) for library preparation and paired-end (2 × 250 bp) sequencing on a MiSeq Illumina instrument (San Diego, CA, USA).

**Data processing**. The MiSeq data were merged with VSEARCH[65] and the sequences were cleaned as follows: sequences were removed if they presented ambiguous bases "N", a length shorter than 200 bp, and had a mismatch in the forward or reverse primers. The putative chimaeras were detected by VSEARCH. The remaining rRNA 18S sequences were clustered into "molecular species" (OTU) at a 95% similarity threshold according to Mangot et al.[66] with VSEARCH (option cluster_small sorted by length). This process was implemented using the pipeline PANAM[67] (https://github.com/panammeb/PANAM2).

Technical HTS replicates were merged after assessing their similarity (Supplementary Fig. 9). For each lake and core depth, reads of replicate samples were summed together and subsampled at their average sequencing depth.

The taxonomic affiliation was performed using the RDP algorithm[68] on OTUs representative sequences with a confidence bootstrap threshold of 75%. The Protist Ribosomal Reference database PR2[69] was used as reference. Taxonomic names were harmonized following the revision to the classification of eukaryotes proposed by Adl et al.[25]. OTUs were classified into taxonomic groups for each of the 12 ranks of this classification (Supplementary Data 1) but given the difficulty in classifying short DNA sequences into low taxonomic ranks, only the three highest ranks were used in the statistical analyses (see the list of taxa in Supplementary Table 5).

In addition, functional trophic groups were attributed to OTUs based on their estimated taxonomy. Most of the information on trophic classification of micro-eukaryotes was obtained from Adl et al.[25] completed by specialized literature available for each taxon (see Adl et al.[25] for references). We divided OTUs into five trophic functional groups: consumers (bacterivores, algivores, and omnivores), saprotrophs, parasites, photosynthetics (exclusively autotrophs), and mixotrophs (consumers with photosynthetic capacity). The taxonomic and functional classification of each OTU is given in Supplementary Data 1.

Rare OTUs with less than ten reads in the complete dataset and OTUs identified as multicellular animals (Metazoa) and terrestrial plants (Embryophyta) were removed. OTUs identified as unclassified Eukaryota were considered unreliable and were excluded from the analyses. The complete tracking of the number of DNA reads and OTUs during the filtering and data pre-processing steps is available as Supplementary Information (Supplementary Fig. 2).

All statistical analyses were conducted with R 3.5.3. Scripts to reproduce the results are available at https://doi.org/10.5281/zenodo.3662243.

**Community analyses**. To control for differences in sequencing depths between samples, count data were converted into relative proportions. This approach was preferred over within-sample variance normalization methods as it produces more accurate and reliable comparisons at the community level[70].

Multivariate analyses were conducted with R and the vegan package[71]. Nonmetric Multidimensional Scaling analysis (NMDS) on Bray–Curtis dissimilarities was used to visualize dissimilarities in OTUs composition among communities in a reduced two-dimensional space.

To test the differences in the community composition between top and bottom, we used a permutational multivariate analysis of variance (PERMANOVA[72]) on Bray–Curtis dissimilarities with 1000 permutations. To take account of the dependency within repeated measures, we included a blocking structure to the permutation design to prevent data shuffling across lakes.

To test the changes in dispersion among communities between top and bottom we used a paired test for homogeneity of multivariate dispersions adapted from the procedure of Anderson[73]. We computed principal coordinate decomposition

(PCoA) of the Bray–Curtis dissimilarity matrix in order to obtain euclidean distances between samples. Then, in each group (top and bottom), we computed the distance of each lake to the center of the group (defined here as its geometric median). Finally, lakes-to-center distances were compared between top and bottom using a paired Wilcoxon test.

In addition, multivariate analyses were conducted on the OTU matrix using the Jaccard dissimilarity index and on the taxa matrix using the Bray–Curtis dissimilarity index (Supplementary Figs. 5 and 6).

We used a regression tree model to investigate the effect of lake elevations on the magnitude of lake community changes, which was measured for each lake by the Bray–Curtis dissimilarities between the top and bottom communities. As two groups of lakes could be clearly identified, a decision tree was preferred over a classical regression to better capture the non-linearity in the data. Splits were determined by variance reduction and the optimal number of splits was determined via cross-validation using the tree fitting algorithm of the R package rpart[74].

We analyzed the differences in abundances of micro-eukaryotes between top and bottom with the DESeq2 framework[75] applied on raw count data. Changes in taxonomic and trophic groups are reported using the logarithmic ($\log_2$) fold change value, which expresses how much the number of reads changes between bottom (the reference) and top. Following the DESeq2 procedure, the difference in abundance between bottom and top was tested for each taxon and trophic group using a Wald test and all $p$-values were adjusted for multiple hypothesis testing using the method of Benjamini and Hochberg.

**Reporting summary**. Further information on research design is available in the Nature Research Reporting Summary linked to this article.

## Data availability
All raw reads are available through ENA (https://www.ebi.ac.uk/ena) using the study accession number PRJEB35411. The taxonomic affiliation was performed using the Protist Ribosomal Reference database (PR2) available at https://pr2-database.org/. Any additional data needed to reproduce the results are available at https://doi.org/10.5281/zenodo.3662243.

## Code availability
The R scripts to reproduce the analyses and results are available at https://doi.org/10.5281/zenodo.3952263.

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

## Acknowledgements

We thank Cecile Chardon (INRAE) for technical support, and Paul MacKeigan, Valentin Vasselon, and Cécilia Barouillet for perceptive comments on this manuscript. This work was supported by Suez Lyonnaise des eaux Project "Outils innovants pour la diagnose écologique et la gestion des lacs". F.K. was supported by the Pole Research & Development on Lacustrine Ecosystems (ECLA) of the French Biodiversity Agency (OFB). We thank OLA (Observatory on LAkes), the information system OLA Data (https://doi.org/10.15454/VBWYWG), the research infrastructure AnaEE-France (Analysis and Experimentation on Ecosystems), and the "Observatoire Hommes-Milieux Pyrénées Haut Vicdessos - Labex DRIIHM ANR" for providing limnological data.

## Author contributions

I.D., L.M., D.E., and D.D. designed research; I.D., L.M., D.E., D.D., F.K., D.G., and D.R. performed research; F.K., D.D., and I.D. analyzed data; and I.D., F.K., L.M., D.E., D.G., D.R., and D.D. wrote the paper.

## Competing interests

The authors declare no competing interests.
