## [Peer Review File · Nature Communications]

Reviewers' Comments:

Reviewer #1:

Remarks to the Author:

As I noted to the Editor when I was asked to review this paper, I do not work directly in the field of sedimentary DNA, but I have been trying to follow the literature. I was asked to assess the submitted manuscript for its relevance to the broad paleolimnological literature. In any event, I am confident in the DNA analyses as the paper includes some of the world leaders in sedimentary DNA. The remaining approaches are well-established and robust paleolimnological protocols.

I believe this is a strong paper with novel results that will be of interest to a very broad readership.

I had two issues of a general nature that I thought could be clarified:

Major comments:

1) My principal questions when reading the paper are about the dating of the 'bottom' samples. My lab has extensively used the top-bottom approach, starting with its first applications in our acid rain work (longer ago than I wish to acknowledge). I agree it is a powerful approach for regional assessments, like this paper addresses. But especially the basal dates are often estimates. Which is fine, but this issue should be acknowledged more I think. It is not clear from either the manuscript nor the supplemental info how old the basal dates are or how much they vary in age, yet this information is critical to the interpretation of the presented trends.

In a similar vein, the manuscript states "The thickness of sediment samples was individually adjusted so that each sample covered 10 to 15 years". How was this done? The dating information in the SI indicates that some of the cores were dated through measurements of stable lead, some were dated using age-depth models of radioisotopes, and others were interpolated from radiocarbon dates? Do radiocarbon dates provide the precision necessary to say a particular interval only spanned 10-15 years. Furthermore, it is not clear how the dating information would allow the subsampling to be restricted to specific time frames on a core-by-core basis? Wouldn't the subsampling have been performed to calculate the dates (e.g. would the terrestrial organic remains used to obtain radiocarbon dates been isolated during subsampling?). If possible, please clarify these issues.

2) I never really got a sense of where the study lakes are located. The only

mention is that they are temperate. But no sense whether these are all or anything else. Could not at least a summary of this be presented in the paper? should be stated ideally in Intro or Methods at very least.

Minor comments:

My comments below are generally are all minor.

Line 26, and I will touch on this later in review as well. You note as one of your main conclusions that remote high-altitude lakes were less impacted than lowland lakes. Later you expand on this by noting the most likely cause is that the lowland lakes have more local impacts, which makes sense. I wonder if you had any "reference" or relatively undisturbed lakes that are lowland and how they compare to similarly undisturbed high-altitude lakes? I bring this up for when we look at say diatoms or chrysophytes or fossil pigments, often we see the biggest changes as we get closer to the poles and up mountains as we are crossing important climate-related thresholds, namely big changes in ice cover and/or changing thermal stratification patterns. The biggest, most sensitive climate-related changes occur in higher polar and higher elevation – where the biggest changes in ice etc are happening. In fact, we have lakes with 100% turnovers in microfossils (see for example review by Rühland et al. (2015, J Paleolimnology) in some of our high Arctic sites with massive changes in declining ice cover.

So what I am saying is what you write makes perfect sense (IF the lowland lakes are affected by local impacts) but is it possible to do a comparison of high altitude undisturbed with low altitude undisturbed lakes? Maybe you do not have this in your dataset, and so that is fine. But it might be worth making it clear you might/probably would get a very different signal if you had these undisturbed low elevation lakes?

Line 34: This is of course just a partial list of the recent impacts – for example, invasive species, etc. Maybe end that sentence with "amongst other stressors" to make clear you acknowledge that is not a full list.

Line 35: Change "since millenia" to "for millenia"

Line 42: it is good to cite the important Lake Superior paper (reference 10) – but this does imply (the way you wrote sentence) that this was only recorded in Lake Superior. In fact, paleolimnologists have been recording increased algal production with accelerate warming (and less ice cover etc) in a wide spectrum

of lakes. Maybe expand the reference to make clear it is recorded in many lakes. My recent invited perspective article used these increases in production as one of my 3 examples, reviewed in:
Smol, J.P. 2019. Under the radar: long-term perspectives on ecological changes in lakes. Proc. Roy. Soc. B. 286: 20190834.
<http://dx.doi.org/10.1098/rspb.2019.0834>

Line 48: I suggest changing word "good" to "prime"

Line 50: Might add fossil pigments to diatoms and chrysophytes, as a far amount of pigment literature is out there now.

Line 68: When you first introduce the top-bottom approach, I (being a paleolimnologist) read and understand what you are doing very well. And you do refer to one of my textbooks. But Nature Communications has a very broad readership. So I would add a bit to that sentence to make it totally clear what you are doing. Something like:
... using a so-called top-bottom paleolimnological approach (20), whereby DNA preserved in surface (i.e. top) sediments representing recent lake assemblages is compared to a sediment sample from pre-1880 (i.e. bottom) sediments, representing reference conditions.

Line 79: Similar to comment I made above, not sure all readers are understanding 'recent' and 'past' sample. How about writing and clarifying: 'recent' (i.e. the top sediment sample) and 'past' (i.e. the bottom, pre-1880 sediment sample"

Figure 1 and elsewhere. I always get a bit nervous when people use organic carbon as a proxy for in-lake production. We now know much of this carbon in some lakes is not from algae etc., but from terrestrial sources etc. Might be worth noting in some way.

Figure 1: What portion of the DNA concentration differences could be attributable to degradation? Supp Info states that the DNA signal is unlikely to be highly altered by fragmentation in recent sediments (ie. <120 years old); however, it is not clear how old the bottom samples are, or what proportion of them may pre-date 1900.

Line 153: Change "associated to" to "associated with"

Line 185: Maybe instead of "corresponds" a better word is "includes" as the Great Acceleration is mainly the 2nd half of 20th century, not since 120 years ago.

Line 214-215: Please see my comments above on line 26 in abstract about high altitude and low altitude lakes. Maybe this is also a place to clarify this, if possible?

Line 239: Please see my comments at line 42 above.

Line 255: We are recording much higher chrysophyte assemblages, especially scaled chrysophytes (e.g. *Synura*) with warming in oligotrophic, low alkalinity lakes. And this is matched with monitoring programs. Were you able to see any of this in your data? Maybe your lakes are mainly more alkaline.

Line 285: You note preservation was poor in 5 sediment cores. Can you expand on this? Any ideas or observations why this was the case?

Lines 292 and 293: 210 and 137 should be superscripted

Lines 380-385: Should not these numerical techniques be referenced?

Interesting paper. Although I managed to type 3 pages of comments, these are all minor suggestions.

John P. Smol
March 10, 2020

Reviewer #2:

Remarks to the Author:

This is very interesting paper where for the first time sed aDNA has been analysed on multiple low vs high latitude lakes to study microbial eukaryotic community and function changes with the onset of the Anthropocene.

I have a few minor recommendations for improvement:

Line 29 Abstract: Why would anthropogenic changes benefit phototrophs in particular? One would then also expect an increase in mixotrophs feeding on

those primary producers. It is more likely that an increase in nutrients causes a shift towards larger celled photosynthetic taxa that require the presence of also larger, more specialised predators. I also placed a comment about this below.

Line 56: Please cite the pioneering lake sed aDNA studies involving microbial eukaryotes from Coolen et al (2004 EPL) and Boere et al. (2009) (Geobiology).

Line 75: The discrimination between ecological functions of past protists from sed aDNA profiling was recently also done for the Arabian Sea (More et al., 2018). That study showed which taxonomic and functional groups were selected by long periods of oxygen minimum zone extension triggered by Monsoon controlled increased upwelling and eutrophication of the photic zone. The Arabian Sea is of course not a lake system but there may be overlapping environmental and biological principles at play that could be discussed in more detail in this paper.

Line 170: Where did this predation take place? Can the authors predict based on the species composition that this occurred in the past water column or could some of this predation have occurred in the oxygenated surface sediments at the time of deposition? How would this have influenced preservation of the DNA from the various contributing functional groups? The latter needs to be at least discussed.

Line 200. See previous comment about the More et al. 2018 study (EPL). This should at least be cited but preferably also discussed. For example, later at line 217 it states that nutrient enrichment must be an important cause for the observed shifts, but it is not discussed properly, not even in line 224 where another ancient DNA study is cited. However, there are also numerous modern protist ecology studies that have developed ecosystem models based on observed community shifts resulting from ongoing eutrophication. This is kind of mentioned but not very well compared with the sed aDNA results. For example, as mentioned above, eutrophication usually results in a shift towards larger celled primary producers such as diatoms and dinoflagellates and that must also result in a community shift towards larger, more specialised predators. Cell size inferred from the paleocommunities and how this fits those established ecosystem models is not discussed. In contrast, oligotrophic conditions usually result in a predominance of picoplankton and hence also a co-presence of smaller types of predators. Is this evident from the older sediments? This type of discussion would strengthen the reliability of the sed aDNA results. For example, one could argue that the recent increase in phytoplankton could be

due to a preferred postburial degradation of the aging phytoplankton DNA. This brings me back to the comment above where I asked if it is known where the predation has occurred in the past (water column or surface sediments)? I do not think that this is the case because of the young age of the sediments but a more in depth discussion is warranted in general.

Reviewer #3:

Remarks to the Author:

The authors present a very interesting question – how eukaryotic microbial communities changed across 48 lakes between 1880 and 2000 (i.e. pre- and post-Great Acceleration). They developed a solid experimental set-up to test this question and found significant changes between past and recent communities. I found the results novel and important, in particular that a strong reduction of spatial beta-diversity was found between recent and past samples and that photosynthetic and mixotrophic eukaryotes appeared to have increased in the recent sediments. I liked the author's use of Adl et al. to classify their eukaryotic community into functional trophic groups. Overall, the paper is clearly written and pleasant to read. Integrating genomics into paleolimnology will contribute to the field's understanding of past and present microbial communities, which are essential in regulating important ecosystem functions like carbon fixation and greenhouse gas emissions.

The authors raised some valid and considerable concerns about ancient sedimentary DNA in their introduction, discussion, and supplementary information. However, I feel like some important caveats were not mentioned in their manuscript. For example, they did not address the question of leaching of microbes from one sediment strata to another, or whether microbes found in older strata were necessarily "dead". I would like to see these limitations acknowledged and discussed in a future version of this paper. There are also a number of other smaller issues that need to be addressed before the paper can be published.

Major comments:

1) While the methodology is clearly thorough and the authors attempted to avoid contamination as much as possible, I wonder if it is ever possible to infer past microbial communities. How can we guarantee that no leaching occurred from top, recent sediments to bottom, past ones? Could past assemblages be

more diverse because different communities from sediments above leached through it with time? You could maybe argue that even if there was hydrological/geophysical variability among lakes that would alter how the sediments changed through time, the same patterns in biodiversity were observed among lakes suggesting that changes observed were indeed due to the communities present at the time, rather than to how they changed with time in the sediment.

2) I wonder how the overview of the sediment records ties in with the rest of the analysis. Are the total carotenoids levels in the past sediment dry enough to suggest that no organisms could survive there now? Could this suggest that the bottom sediments are only comprised of "dead" microbes and are thus revelatory of ancient communities? This could help reject the idea that the DNA sequenced in the past sediment is instead active DNA of live organisms that can live at the depth of the past sediment. It would need to be backed up with a reference. You could also maybe mention that if you did shotgun sequencing, you could potentially look at damage on the end of your reads to assess whether they are old or not.

Minor comments:

L34: Change "resources" to "resource"

L38: Change "lakes" to "lake"

L38: What type of associated ecological functions?

L42-44: Please add references.

L65: Change "in the end" to "at the end"

L89: Change to "the robustness of the DNA results"

L93-95 and Fig. 1: I'm not sure what to make of these results. Are these differences between past and recent proxies for organic and biogenic compounds due to different conditions in the past/recent sediments or are they simply due to the effect of time on the past sediment? Are there models to suggest the sediments would have evolved that way with time or instead to counter the effect of time on these compounds?

L105: Add “,” after “...Supplementary Fig. 1)”

L112-114: It would be nice to visualize these numbers, perhaps with a Venn diagram in the Supplementary Information.

L130-132: Please add the results of your PERMANOVA as a table in the Supplementary Information.

L131: Change “OTUs” to “OTU”

L153: Change “associated to” to “associated with”

L153-154: “(i.e. a fine taxonomic level for which DNA modification during aging in sediments may be questioned)” <- This statement is unclear, please clarify.

L161-162: Change “obligatory” to “obligate” and add “i.e. photosynthetics” to be consistent with the terminology used in Fig. 4.

L174: Change “which” to “whose”

L195: You are suggesting that microbial communities are inextinguishable. If that’s the case, would that mean that the communities found in the past sediments are also alive? How could we be sure that these represent a past community and not movement of live microbes through the sediment?

L216-217: Is it really true that that high altitude lakes are assumed to be less exposed to local anthropogenic pressures? I have doubts about this statement, please give a more recent reference to support the idea.

L217-218: This conclusion seems to be drawn a little quickly, especially considering that there is no data in this paper to support the fact that local human activities are the primary drivers of this temporal turnover (apart from the dating of the sediment, there is no measure of human activity in the lakes). If you are referring to TOC/pigment analysis to support this, please explain your reasoning and make it clearer. You could otherwise perhaps complement this statement with more data/analysis if you have any related to human activity.

L243: How would you confirm that? Isn’t there literature on this already?

L288: Please be consistent throughout the manuscript and supplementary

information and choose to use either “subsampling” or “sub-sampling”

L294-295: Please give a range for the depths at which the cores were collected.

L.291-294: If the particle sizes of these elements (Pb, Cs) are comparable to microbial eukaryotes, could you use that as a calibration for leaching of microbes from one layer to another? Otherwise could pollen data help with calibration (if available)?

L379: Shouldn't DESeq2 be used on normalized data rather than raw count data (for example with the DESeq2 variance stabilizing transformation)?

Fig.1: I am not sure why the number of lakes isn't equal to 48 and varies among the different measurements taken (A, B and C). Please clarify.

Supplementary information:

Fig.1: Is this from 48 samples extracted in duplicates? Please indicate average sequencing raw data read depth per sample.

Fig. 3 and Fig. 4: Please add stress values in legends for NMDS plots.

Manuscript NCOMMS-20-07936

Assessing the response of micro-eukaryotic diversity to the Great Acceleration: a paleolimnological view based on sedimentary DNA. F. Keck, L. Millet, D. Debroas, D. Etienne, D. Galop, D. Rius, I. Domaizon*

Point-by-point responses to reviewers' comments

Reviewer #1

As I noted to the Editor when I was asked to review this paper, I do not work directly in the field of sedimentary DNA, but I have been trying to follow the literature. I was asked to assess the submitted manuscript for its relevance to the broad paleolimnological literature. In any event, I am confident in the DNA analyses as the paper includes some of the world leaders in sedimentary DNA. The remaining approaches are well-established and robust paleolimnological protocols. I believe this is a strong paper with novel results that will be of interest to a very broad readership.

I had two issues of a general nature that I thought could be clarified:

Major comments:

- **1.1** My principal questions when reading the paper are about the dating of the 'bottom' samples. My lab has extensively used the top-bottom approach, starting with its first applications in our acid rain work (longer ago than I wish to acknowledge). I agree it is a powerful approach for regional assessments, like this paper addresses. But especially the basal dates are often estimates. Which is fine, but this issue should be acknowledged more I think. It is not clear from either the manuscript nor the supplemental info how old the basal dates are or how much they vary in age, yet this information is critical to the interpretation of the presented trends.

In a similar vein, the manuscript states "The thickness of sediment samples was individually adjusted so that each sample covered 10 to 15 years". How was this done? The dating information in the SI indicates that some of the cores were dated through measurements of stable lead, some were dated using age-depth models of radioisotopes, and others were interpolated from radiocarbon dates? Do radiocarbon dates provide the precision necessary to say a particular interval only spanned 10-15 years. Furthermore, it is not clear how the dating information would allow the subsampling to be restricted to specific time frames on a core-by-core basis? Wouldn't the subsampling have been performed to calculate the dates (e.g. would the terrestrial organic remains used to obtain radiocarbon dates been isolated during subsampling?). If possible, please clarify these issues.

We acknowledge that the first submitted version of the manuscript was lacking details about the dating methods used to estimate the age of the samples, in particular the bottom samples. We

edited both the manuscript and the supplementary information to improve clarity. In particular the SI Methods S2 was significantly completed with information about the dating methods. We also added a new table to the SI (SI Table 2) which indicates for each lake, the data and method used to determine the depth and thickness used for sampling. More specifically regarding your questions:

basal dates :

The addition of information in SI Methods S2 and the new SI Table 2 do address this point. To sum up, we reminded, to paleo and non-paleo readerships, that the 19th century is a challenging period to date, because of methodological limitations. However, we ensured that all our bottom samples are anterior to the Great Acceleration and most likely from the second half of the XIXth century, by using either radiocarbon, ^{210}Pb , stratigraphic markers (lead) or a combination of those methods (see SI Table 2).

Thickness of sediment samples :

We modified this sentence to "...at least 10 to 15 years...". Indeed, our objective was not to reach such a level of precision, but primarily to smooth interannual variability in order to get an average biodiversity for both recent (top) and reference (bottom) samples and enable a more robust comparison between lakes.

Precision of radiocarbon dates

Reaching this level of precision was not our objective and we reformulated it accordingly. Of course, radiocarbon dates do not provide this level of precision. In some altitude lakes (SI Table 1), where sedimentation rates are usually low, radiocarbon-based, multi-centennial depth-age models were used to 1) date 19th century sediments and 2) estimate sedimentation rate so that DNA analysis reflects an average biodiversity at decadal scale, hence overcoming interannual variability.

subsampling and dating:

In the corrected SI Methods S2 section, we described much more thoroughly how cores were processed for sampling, subsampling, core logging, and subsequent analyses. In particular, we made it clear that cores were not extruded on the field. Water-sediment interface was stabilized with floral foam, cores were brought to the lab, split into two halves, one for multi-proxies analysis, including DNA, and one for core-logging and dating. Indeed, extruding on the field constrain all the analyses, including chronological ones, to be made on the same subsamples. It also prevents core logging (no Xrf core-scanner analysis possible for instance) and any subsequent change in subsampling (resolution of analysis can only be downgraded by pooling subsamples but not

upgraded). This is the reason why we performed subsampling for chronological purposes (i.e. Xrf core-logging, ^{14}C and $^{210}\text{Pb}/^{137}\text{Cs}$) and proxies analyses on two different halves of the same core.

- **1.2** I never really got a sense of where the study lakes are located. The only mention is that they are temperate. But no sense whether these are all or anything else. Could not at least a summary of this be presented in the paper? should be stated ideally in Intro or Methods at very least.

These information were indeed not explicitly mentioned in the text, the Table S1 presented the coordinates for each lakes, and, in the revised manuscript, we have now (i) included a map (in the Supplementary Information) showing the location of the 48 lakes in France, and (ii) clarified this in the introduction.

Minor comments:

My comments below are generally are all minor.

- **1.3** Line 26, and I will touch on this later in review as well. You note as one of your main conclusions that remote high-altitude lakes were less impacted than lowland lakes. Later you expand on this by noting the most likely cause is that the lowland lakes have more local impacts, which makes sense. I wonder if you had any “reference” or relatively undisturbed lakes that are lowland and how they compare to similarly undisturbed high-altitude lakes? I bring this up for when we look at say diatoms or chrysophytes or fossil pigments, often we see the biggest changes as we get closer to the poles and up mountains as we are crossing important climate-related thresholds, namely big changes in ice cover and/or changing thermal stratification patterns. The biggest, most sensitive climate-related changes occur in higher polar and higher elevation – where the biggest changes in ice etc are happening. In fact, we have lakes with 100% turnovers in microfossils (see for example review by Rühland et al. (2015, J Paleolimnology) in some of our high Arctic sites with massive changes in declining ice cover.

So what I am saying is what you write makes perfect sense (IF the lowland lakes are affected by local impacts) but is it possible to do a comparison of high altitude undisturbed with low altitude undisturbed lakes? Maybe you do not have this in your dataset, and so that is fine. But it might be worth making it clear you might/probably would get a very different signal if you had these undisturbed low elevation lakes?

We agree that it would be interesting to compare the changes between undisturbed high-altitude lakes and undisturbed lowland lakes. Unfortunately we do not have lowland lakes in our dataset that could be considered “undisturbed” by local human pressures. Our set of lakes reflect closely

the situation in Europe, where it is difficult to find, in lowlands, reference sites which are not disturbed by agriculture or urbanization. The situation is likely different in the high Arctic region where local human activities are globally more limited, even at low altitude. There is obviously a strong gradient of human pressures in our data that is highly correlated with altitude and we provide additional data in the revised version of the manuscript to support this statement (see also our response to #3.20). Therefore it is not possible for us to decorrelate the specific effect of climate-related changes and how the altitude could exacerbate it.

Nonetheless, we readily acknowledge that if our lakes were similarly undisturbed across the altitude gradient, we could get a very different picture. We slightly modified the abstract to insist on the fact that altitude is used and interpreted here as a proxy for human imprint (for this set of lakes). Additionally we edited the Discussion as you suggested in #1.15.

See L 200-2004 of the revised manuscript.

- **1.4** Line 34: This is of course just a partial list of the recent impacts – for example, invasive species, etc. Maybe end that sentence with “amongst other stressors” to make clear you acknowledge that is not a full list.

We added “amongst other stressors” to the text.

- **1.5** Line 35: Change "since millenia" to "for millenia"

We changed the text.

- **1.6** Line 42: it is good to cite the important Lake Superior paper (reference 10) – but this does imply (the way you wrote sentence) that this was only recorded in Lake Superior. In fact, paleolimnologists have been recording increased algal production with accelerate warming (and less ice cover etc) in a wide spectrum of lakes. Maybe expand the reference to make clear it is recorded in many lakes. My recent invited perspective article used these increases in production as one of my 3 examples, reviewed in:

Smol, J.P. 2019. Under the radar: long-term perspectives on ecological changes in lakes. Proc. Roy. Soc. B. 286: 20190834. <http://dx.doi.org/10.1098/rspb.2019.0834>

We changed the text and added the reference. This was indeed lacking.

- **1.7** Line 48: I suggest changing word “good” to “prime”

Done

- **1.8** Line 50: Might add fossil pigments to diatoms and chrysophytes, as a far amount of pigment literature is out there now.

We agree and added fossil pigments.

- **1.9** Line 68: When you first introduce the top-bottom approach, I (being a paleolimnologist) read and understand what you are doing very well. And you do refer to one of my textbooks. But Nature Communications has a very broad readership. So I would add a bit to that sentence to make it totally clear what you are doing. Something like:

... using a so-called top-bottom paleolimnological approach (20), whereby DNA preserved in surface (i.e. top) sediments representing recent lake assemblages is compared to a sediment sample from pre-1880 (i.e. bottom) sediments, representing reference conditions.

We agree with your remark and modified the text accordingly.

- **1.10** Line 79: Similar to comment I made above, not sure all readers are understanding ‘recent’ and ‘past’ sample. How about writing and clarifying: ‘recent’ (i.e. the top sediment sample) and ‘past’ (i.e. the bottom, pre-1880 sediment sample”

We edited as recommended.

- **1.11** Figure 1 and elsewhere. I always get a bit nervous when people use organic carbon as a proxy for in-lake production. We now know much of this carbon in some lakes is not from algae etc., but from terrestrial sources etc. Might be worth noting in some way.

We agree and added a few words in the discussion to stress this limit.

See L 227-228 of the revised manuscript.

- **1.12** Figure 1: What portion of the DNA concentration differences could be attributable to degradation? Supp Info states that the DNA signal is unlikely to be highly altered by fragmentation in recent sediments (ie. <120 years old); however, it is not clear how old the bottom samples are, or what proportion of them may pre-date 1900.

It is hard to answer this question because most studies about DNA degradation focused on signal quality (fragmentation/mutation) and not on quantity of total DNA. As explained we avoided the uppermost sediment layers because the first few years after deposition are critical due to the biological activity at the surface sediment. It seems from previous data we collected for annually laminated sediments (e.g. Capo et al 2017, other unpublished data) that (after post-depositional

loss in surface sediments) the quantity of total DNA is stable during the first 25 years of burying, then we detected a loss of DNA concentration (at least with the method we used); this loss could reach up to 15% of the DNA material (unpublished data); however those observation are not directly transposable to all lakes. We really think that the quantity of total DNA material preserved in sedimentary archives could be a relevant paleoproxy (of global productivity for instance); however further studies are required to explore the links between this potential indicator and other proxies and to consolidate its interpretation.

In any case, in the time interval studied here, it is unlikely that the DNA signal was highly altered, as suggested by several studies that report good preservation of DNA for hundreds or even thousands of years. But we cannot exclude that a small portion of the DNA concentration differences between Top and Bottom (Figure 1) could be attributable to degradation.

We have added a sentence to specify this in the revised manuscript. Please see Supplemental text Method S3 and, in the revised manuscript L100-102.

In the Supplementary Information, our sentence was unclear. Our message was that given the time scale investigated in this study, it is unlikely that the DNA signal could be significantly affected by degradation. We clarified this sentence. Additionally and following your comment #1.1 we provide more information regarding the age of the bottom samples (Supplementary Information).

- **1.13** Line 153: Change "associated to" to "associated with"

Done in the revised version

- **1.14** Line 185: Maybe instead of "corresponds" a better word is "includes" as the Great Acceleration is mainly the 2nd half of 20th century, not since 120 years ago.

We agree and changed to "includes".

- **1.15** Line 214-215: Please see my comments above on line 26 in abstract about high altitude and low altitude lakes. Maybe this is also a place to clarify this, if possible?

Please see our detailed response to your comment #1.3. We added a sentence to make it clear that our conclusions do not contradict the fact that high altitude lakes can actually respond more strongly to climate-related changes than lowland lakes, but this cannot be directly tested with our data.

- **1.16** Line 239: Please see my comments at line 42 above.

We added Smol 2019 to the references cited for the increase in primary production.

- **1.17** Line 255: We are recording much higher chrysophyte assemblages, especially scaled chrysophytes (e.g. *Synura*) with warming in oligotrophic, low alkalinity lakes. And this is matched with monitoring programs. Were you able to see any of this in your data? Maybe your lakes are mainly more alkaline.

This comment is interesting, we indeed observe from our data that Synurales responded positively to recent environmental conditions (according to DESEQ2 analysis). Synurales are however not dominant in terms of DNA reads, and Chrysophytes also include colorless/unpigmented groups (e.g. *paraphysomonas*, *spumella*) for which the temporal trend is different.

Because in this manuscript we presented data aggregated at the 3rd taxonomy rank of Adl et al, the dynamics of Synurales was not particularly brought out (in Fig S8 where Top-Bottom changes are presented for taxonomic group, Synurales are included in Ochrophyta group to which Chrysophytes belong to).

It is also noticeable that a complete paleo-reconstruction of micro-eukaryotic diversity based on sed-DNA has previously shown that, for 2 lakes (lake Bourget and Annecy), changes in phosphorus concentrations and recent warming have acted synergistically to promote Chrysophyceae and Dinophyceae over the last century (Capo et al 2017).

Additionally, the observed trend for Chrysophytes (increase in recent periods) also matches with data from monitoring programs. The survey of water quality done in few calcareous lakes (e.g. Léman, Aiguebelette) shows an increase of Chrysophytes (*Synura* or *Dinobryon*) since the 80s (results extracted from French technical reports produced by observatory on lakes OLA which is described in Rimet et al 2020).

We propose to add a sentence in the supplemental text to mention this. Please see Supplemental text Methods S1 (p3-4)

Bibliographic references :

- Capo et al 2017 *Environmental Microbiology*, 19(7):2873-2892 DOI: 10.1111/1462-2920.13815
- Rimet et al 2020. *Journal of Limnology*. 10.4081/jlimnol.2020.1944.

- **1.18** Line 285: You note preservation was poor in 5 sediment cores. Can you expand on this? Any ideas or observations why this was the case?

For 5 sediment cores (table sup 1), though the DNA concentrations for top and bottom strata were not particularly low, we had difficulties in amplifying a sufficient amount of the targeted barcode (eukaryotes). It is noticeable that cyanobacterial DNA (prokaryotes) was amplified for most of these DNA samples (unpublished data), showing that other biological material was preserved and accessible for DNA analysis.

Our hypothesis is that, in those sed-DNA extracts, eukaryotic DNA was in very low quantity and could be more degraded than in other lakes. Through sedimentation, a part of the eukaryotic sedimenting material can be lost (Capo et al 2015) ; after deposition on sediment surface, we know from bibliography and previous experiments that different physical and chemical sediment characteristics (temperature, oxygen level, etc) might have an impact on the DNA preservation (e.g. Coolen & Gibson 2009 ; Parducci et al 2017, 2019 ; Torti et al 2015, ...). In organic rich lake sediments, high activity of microbial communities in the oxygenated surface sediments is probably the biggest challenge for the survival of sed aDNA. Through diagenetic processes, DNA can also be damaged, with DNA strand breakage, miscoding lesions and DNA crosslinks, that may lead to difficulty for the analysis of the molecular signal (Torti et al 2015, Pedersen et al 2015; Willerslev and Cooper 2004). The use of short DNA barcode generally allows to circumvent this problem; however in this study we used a 260bp long barcode which is not extremely short ; this choice has been made to optimize the taxonomic identification for micro-eukaryotes and sed-DNA (Capo et al 2016) ; however we can not exclude that shorter barcodes would be more efficient.

The 5 lakes have contrasted trophic status (in 2015) from eutrophic to oligotrophic, and very different morphology with depth from ~8m to 40m. Our knowledge about these 5 lakes is not sufficient to decipher which parameter (loss during sedimentation through the water column, temperature and oxygen conditions at the bottom of the lake, bioturbation, bacterial degradation occurring at the subsurface of sediment) explain the most this low preservation of eukaryotic DNA material.

Bibliographic references :

- Coolen & Gibson 2009 PAGES News. 17. 104-106. [10.22498/pages.17.3.104](https://doi.org/10.22498/pages.17.3.104).
- Capo et al 2016 Molecular Ecology, 25(23):5925-5943. DOI: [10.1111/mec.13893](https://doi.org/10.1111/mec.13893)
- Parducci et al 2017, New Phytologist (2017) 214: 924–942. doi: [10.1111/nph.14470](https://doi.org/10.1111/nph.14470)
- Pedersen et al 2015 Philos Trans R Soc Lond B Biol Sci. 370(1660) doi: [10.1098/rstb.2013.0383](https://doi.org/10.1098/rstb.2013.0383)
- Torti et al 2015 Marine genomics 24:185-196 doi: [10.1016/j.margen.2015.08.007](https://doi.org/10.1016/j.margen.2015.08.007).
- Willerslev and Cooper 2004 Proc. R. Soc. B.2723–16 <https://doi.org/10.1098/rspb.2004.2813>

- **1.19** Lines 292 and 293: 210 and 137 should be superscripted

Done in the revised version

- **1.20** Lines 380-385: Should not these numerical techniques be referenced?

Both the Wald test and Benjamini Hochberg correction are part of the implementation of the DESeq2 framework which is referenced a few lines before. We edited the text to make this point

clearer. Since both are standard methods and are referenced in the DESeq2 paper, we think it will be straightforward for the interested reader to find them.

Reviewer #2

This is very interesting paper where for the first time sed aDNA has been analysed on multiple low vs high latitude lakes to study microbial eukaryotic community and function changes with the onset of the Anthropocene.

I have a few minor recommendations for improvement:

- **2.1** Line 29 Abstract: Why would anthropogenic changes benefit phototrophs in particular? One would then also expect an increase in mixotrophs feeding on those primary producers. It is more likely that an increase in nutrients causes a shift towards larger celled photosynthetic taxa that require the presence of also larger, more specialised predators. I also placed a comment about this below.

We agree that anthropogenic changes could benefit both phototrophs and mixotrophs. We rephrased the sentence to clarify this point (see L29).

Some mixotrophs may benefit from an increased planktonic production as they can feed on small size algae (picoplankton). This point is consistent with our results.

We also keep in mind that mixotrophic protists actually present a spectrum of nutritional strategies (e.g. four groups based on their nutritional behaviour according to Jones al 2000) but this is a bit too complex to be developed in the abstract. Finally, we agree that the effect of environmental changes on cell size in communities is very interesting but see our answer to #2.5

Bibliographic references :

-Jones et al 2000 *Freshwater Biology*, 45(2) 219-226 <https://doi.org/10.1046/j.1365-2427.2000.00672.x>

- **2.2** Line 56: Please cite the pioneering lake sed aDNA studies involving microbial eukaryotes from Coolen et al (2004 *EPSL*) and Boere et al. (2009) (*Geobiology*).

We added these two papers (L58).

- **2.3** Line 75: The discrimination between ecological functions of past protists from sed aDNA profiling was recently also done for the Arabian Sea (More et al., 2018). That study showed which taxonomic and functional groups were selected by long periods of oxygen minimum zone

extension triggered by Monsoon controlled increased upwelling and eutrophication of the photic zone. The Arabian Sea is of course not a lake system but there may be overlapping environmental and biological principles at play that could be discussed in more detail in this paper.

Thank you for suggesting this paper (More et al. 2018). We agree it is important to acknowledge the existence of previous studies that used ancient DNA to show temporal shifts in functional diversity in relation to changes in environmental conditions. As you wrote, there are obvious differences between the Arabian sea and our set of lakes but there are also interesting parallels to be drawn between these studies. We therefore included a sentence in the discussion to point this out.

See L209-213 of the revised manuscript.

- **2.4** Line 170: Where did this predation take place? Can the authors predict based on the species composition that this occurred in the past water column or could some of this predation have occurred in the oxygenated surface sediments at the time of deposition? How would this have influenced preservation of the DNA from the various contributing functional groups? The latter needs to be at least discussed.

The phagotrophic Cercozoans (Glissomonadida, Paracercomonadida, Thecofilosea, Cercomonadida) detected here are both constituents of pelagic and benthic food webs. However a large part of the phagotrophic cercozoans we detected here are gliding biflagellates as the Cercomonads (e.g. *Paracercomonas*) and Glissomonads (e.g. *Sandona* and *Allapsa*). They generally glide over the substrate and should be most of the time associated with a physical surface. These groups are thus more particularly found in benthic areas, at the subsurface of sediment. For instance Glissomonads are known to be constituents of soil protistan fauna (Howe et al 2019) and were also observed in lakes (both in benthic and periphytic samples ; Prokina & Philippov 2018 ; in the 5 first cm of sediment in lake baikal, Hi et al 2019).

However, even though most species of these heterotrophic groups are temporarily or permanently attached to substrate or glide on its surface, there is a considerably large number of species detected in plankton, and many heterotrophic flagellates have swimming forms, which can be registered in the plankton as well (and be active grazers in the water column).

Though it would be a bit risky to state that the predation has occurred more particularly in one of the two compartments (water or surface sediment), we find extremely relevant to mention that the presence and activity of phagotrophic micro-eukaryotes in the surface sediments (at the time of deposition) is of interest for the interpretation of paleorecords.

The protistan grazing activity (at oxygenated surface) could have influenced the DNA signal in a positive way by reducing the amount of bacteria able to decompose dead material organic matter; however because in oxygenated surface sediments both dead cells and living cells can be preyed

upon by the active micro- meio- and macro-fauna, the consequence of the grazing activity is more probably a reduction of the total amount of organic material that could be buried in sediment at this time.

We also see the interest of tracking the presence/absence (and relative abundance) of these taxa associated to oxygenated conditions at the surface sediment, to possibly infer the temporal variation in the level of hypoxia at the bottom of the lake (shift from oxygenated to anoxic bottom) which is of great interest in terms of functioning of the lakes (though we already have this type of information with the taxonomic identification of chironomids remains).

We added some information in the revised manuscript : see L272-276 (discussion) and L156 (results).

Bibliographic references :

- Hi et al 2019 FEMS Microbiology Ecology, 93, 2017, fix073
- Howe et al 2019 Protist, Vol. 160, 159–189, May 2009
- Prokina & Philippov 2018 Protistology 12 (2), 81–96 (2018)

● **2.5** Line 200. See previous comment about the More et al. 2018 study (EPSL). This should at least be cited but preferably also discussed. For example, later at line 217 it states that nutrient enrichment must be an important cause for the observed shifts, but it is not discussed properly, not even in line 224 where another ancient DNA study is cited. However, there are also numerous modern protist ecology studies that have developed ecosystem models based on observed community shifts resulting from ongoing eutrophication. This is kind of mentioned but not very well compared with the sed aDNA results. For example, as mentioned above, eutrophication usually results in a shift towards larger celled primary producers such as diatoms and dinoflagellates and that must also result in a community shift towards larger, more specialised predators. Cell size inferred from the paleocommunities and how this fits those established ecosystem models is not discussed. In contrast, oligotrophic conditions usually result in a predominance of picoplankton and hence also a co-presence of smaller types of predators. Is this evident from the older sediments? This type of discussion would strengthen the reliability of the sed aDNA results. For example, one could argue that the recent increase in phytoplankton could be due to a preferred postburial degradation of the aging phytoplankton DNA. This brings me back to the comment above where I asked if it is known where the predation has occurred in the past (water column or surface sediments)? I do not think that this is the case because of the young age of the sediments but a more in depth discussion is warranted in general.

As recommended, we tried to include additional information in the discussion section (and in the supplemental information) to point out the consistency of our results with established knowledge regarding the effects of nutrients enrichment on (phyto)planktonic groups.

For a substantial part of the OTUs we detected here, it is risky to infer the size of taxa, due to the limited taxonomic resolution we reached. It's why our discussion is not specifically focused on the size structure of phytoplanktonic assemblages, but more generally on the composition and structure of the phytoplankton community that could indeed be used as a biological indicator for eutrophication.

In lake ecosystems, in particular cyanobacteria are favored by eutrophication, and also euglenids and some groups among the chlorophytes are also associated with eutrophication. Known trophic status indexes therefore use typical algal groups of eutrophic conditions (Cyanobacterial, Chlorophyceae and euglenophyceae (for hyper eutrophic water). In Oligotrophic conditions some Zygnemataceae (Desmids) are known to be well represented; while a dominance of Dinophyceae and Cryptophyceae is generally associated to mesotrophic conditions (Stockner 1972; Laplace & Feret 2016 ; Engstrom et al 2006). This corresponds of course to a very general view of the relationships between phytoplanktonic composition and lake trophic status.

Our data are in line with this general trends, the DESEQ2 analysis showed for instance that within Chlorophyta, Chlorophyceae tend to increase in recent periods while Zygnemataceae (including Desmidis taxa as Closterium) tend to decrease in recent periods, which is consistent with changes in nutrients conditions.

Modifications in the revised manuscript are visible L239-246 and in the supplemental text Method S1 (end of p3 & beginning of p4: reference to the comparison with traditional monitoring data).

Bibliographic references :

Engstrom et al 2006 *Ecological Applications*, 16(3), 2006, pp. 1194–1206

Laplace-Tretyure & Feret. 2016 *Ecological Indicators*, Elsevier,2016,69,pp.686-698.10.1016/j.ecolind.2016.05.02

Stockner J. 1972 *Verhandlungen des Internationalen Verein Limnologie*. 18:1018-1030

Reviewer #3

The authors present a very interesting question – how eukaryotic microbial communities changed across 48 lakes between 1880 and 2000 (i.e. pre- and post-Great Acceleration). They developed a solid experimental set-up to test this question and found significant changes between past and recent communities. I found the results novel and important, in particular that a strong reduction of spatial beta-diversity was found between recent and past samples and that photosynthetic and mixotrophic eukaryotes appeared to have increased in the recent sediments. I liked the author's use of Adl et al. to classify their eukaryotic community into functional trophic groups. Overall, the paper is clearly written and pleasant to read. Integrating genomics into paleolimnology will contribute to the field's understanding of past and present microbial communities, which are

essential in regulating important ecosystem functions like carbon fixation and greenhouse gas emissions.

The authors raised some valid and considerable concerns about ancient sedimentary DNA in their introduction, discussion, and supplementary information. However, I feel like some important caveats were not mentioned in their manuscript. For example, they did not address the question of leaching of microbes from one sediment strata to another, or whether microbes found in older strata were necessarily “dead”. I would like to see these limitations acknowledged and discussed in a future version of this paper. There are also a number of other smaller issues that need to be addressed before the paper can be published.

Major comments:

- **3.1** While the methodology is clearly thorough and the authors attempted to avoid contamination as much as possible, I wonder if it is ever possible to infer past microbial communities. How can we guarantee that no leaching occurred from top, recent sediments to bottom, past ones? Could past assemblages be more diverse because different communities from sediments above leached through it with time? You could maybe argue that even if there was hydrological/geophysical variability among lakes that would alter how the sediments changed through time, the same patterns in biodiversity were observed among lakes suggesting that changes observed were indeed due to the communities present at the time, rather than to how they changed with time in the sediment.

Though vertical migration or leaching of DNA has been observed in sediments of caves or soils (uncompacted sediments), DNA leaching in lake sediments is thought to be unlikely and sed-DNA is thought to allow accurate temporal reconstruction (Anderson-Carpenter et al. 2011; Ficetola et al 2018, Sjögren et al 2017).

As reported by Anderson-Carpenter et al, periodic downward percolation of water can move DNA in porous, granular sediments of caves and soil profiles; lake sediments, in contrast, are permanently saturated and gravitational percolation of water does not occur. Once sediments are compacted, vertical advection of pore water is minimal, and multivalent metals and organic compounds (pigments, organic molecules with more than 15 carbon atoms) are immobilized in the sediment matrix. Large organic molecules such as DNA are likely to adhere to solid-phase sediments (particles, particulate organic matter). Organic pollutant as PAHs & PCBs molecules have shown relevant patterns in lake sediments, with increase of these molecules at locally appropriate time-horizons, with no evidence of vertical leaching (Smol 2008; MeyersPA 2003) . The most compelling evidence against DNA leaching in sediments comes from the numerous records that show temporal congruence between paired DNA and morphological (subfossil) or

geochemical records (e.g. Coolen et al. 2006; Stoof- Leichsenring et al. 2012; Epp et al. 2015; Belle et al 2016 ; Dulias et al 2016 ; Tse et al 2018) and between sed-DNA and monitoring data (water column long-term survey) (e.g. Savichtcheva et al. 2014; Monchamp et al 2016). In our particular case, for some lakes we monitored (Observatory on French Lakes), we compared the temporal changes detected from sedDNA (data from this manuscript, but also data previously published) for cyanobacteria (e.g. Savichtcheva et al. 2011 & 2014) and for eukaryotes. For instance we verified the consistency between the changes in the relative proportion of Chrysophytes detected by sedDNA with data from monitoring programs done in lakes Léman and Aiguebelette (increase of Chrysophytes since the 80s in those lakes). This is however obviously not possible to perform such comparison for all the taxonomic groups (or all the lakes) due to the lack of long-term monitoring data.

Moreover, if DNA was significantly leaching from top to bottom sediment it would result (in our data) in an homogenization of the diversity between the two strata while we precisely observe the opposite. As you noted, despite the different lake typologies (with hydrological and geophysical variability that could lead to differences in paleogenetic signal preservation), the global patterns we found in our data suggest a true difference between Top and Bottom communities.

We cannot completely exclude the presence of active cells in compacted sediments; however, if we exclude resting cells considered as dormant but revivifiable (Ellegard et al 2020), and the taxa living at the surface of sediment (benthic protists in oxygenated zones), then, the rest of potentially active eukaryotic cells are most probably rare extremophile specialists representing a negligible fraction of the immense diversity of the total micro-eukaryote community. None of the major groups we identified to explain the changes between Top and Bottom samples are known to live in deep sediments.

We added a short paragraph in the Supplemental Information (section Methods S1 page 4 of the pdf), in order to give some of the bibliographic references we mention here.

Bibliographic references:

- Anderson-Carpenter et al. 2011 BMC Evolutionary Biology 11:30 <http://www.biomedcentral.com/1471-2148/11/30>
- Belle et al 2016 Quaternary Science Reviews <https://doi.org/10.1016/j.quascirev.2016.02.019>
- Coolen et al 2006 NATO Science Series: IV-Earth and Environmental Sciences, Neretin, L. N., Springer, Dordrecht, 41–65.
- Dulias et al 2016 J Paleolimnol 57, 51–66 (2017). <https://doi.org/10.1007/s10933-016-9926-y>
- Ellegard et al 2020 Communications biology <https://doi.org/10.1038/s42003-020-0899-z>
- Epp et al 2015 Quaternary Science Reviews 117: 152–163.
- Ficetola et al 2018 Science Advances 4, 5, eaar4292 DOI: 10.1126/sciadv.aar4292
- Haile J, et al 2007 Mol Biol Evol 24:982–989
- Inagaki et al. 2005 Astrobiology 10.1089/ast.2005.5.141
- MeyersPA 2003 Organic Geochemistry 2003, 34:261-289.
- Monchamp et al 2016 Applied Env Microbiol 2016 10.1128/AEM.02174-16

- Savichtcheva et al. 2011 *Applied Env. Microbiol.* 77(24), p.8744-8752.
- Savitcheva et al 2014 *Freshwater Biology.* 60, 31–49. doi:10.1111/fwb.12465.
- Sjögren et al 2017 *New Phytologist* <https://doi.org/10.1111/nph.14199>
- Stoof-Leichsenring et al 2012 *Mol. Ecol.*, 21, 1918–1930
- Smol JP 2008 *Pollution of lakes and rivers: a paleoenvironmental perspective* Oxford: Blackwell Publishing; 2008.
- Tse et al 2018 *Environ. Sci. Technol.*, 52, 12, 6842-6853 <https://doi.org/10.1021/acs.est.7b06386>

● **3.2** I wonder how the overview of the sediment records ties in with the rest of the analysis. Are the total carotenoids levels in the past sediment dry enough to suggest that no organisms could survive there now? Could this suggest that the bottom sediments are only comprised of “dead” microbes and are thus revelatory of ancient communities? This could help reject the idea that the DNA sequenced in the past sediment is instead active DNA of live organisms that can live at the depth of the past sediment. It would need to be backed up with a reference. You could also maybe mention that if you did shotgun sequencing, you could potentially look at damage on the end of your reads to assess whether they are old or not.

Are the total carotenoids levels in the past sediment dry enough:

We are not sure what you mean exactly by dry enough, but as mentioned before lacustrine are well compacted, they are however wet, we generally always estimate the % of humidity to express the different variables/metrics per g of dry sediment. Prokaryotes, and in particular Archaea can live in deep sediments.

Are bottom sediments only comprised of “dead” microbes:

There are actually dormant living forms in lake sediments (resting stages, eggs, spores, cysts...). These propagules can constitute a significant source of DNA. There is no good reason to exclude this DNA and use only free DNA and DNA from dead cells. Indeed, propagules are relevant paleo-indicators, as long as their position remains stable in the sediment (they are not actively moving along the core).

Ellegaard et al (2020) recently published a review about the use of DNA (that can be preserved in sediments both in bulk sediment and in intact, viable resting stages), showing the great potential for the combined use of ancient eDNA and resurrected long-term dormant organisms, to reconstruct trophic interactions and evolutionary adaptation to changing environments.

The micro-eukaryotic taxa we detected are mainly known as planktonic groups ; some (heterotrophs) that can live at the surface of oxygenated sediment (as reported in the revised manuscript; e.g L270-276) were also found, but these taxa do not survive in hypoxic/anoxic conditions.

As we explained in #3.2, it is very unlikely that a significant part of the eukaryotic diversity we detected here actually corresponds to active taxa that move freely between sediment layers.

Shotgun sequencing to look at DNA damages :

We added a sentence (supplemental text Methods S1 p.4) to mention that shotgun sequencing allows to differentiate ancient DNA that has been damaged (% of AT at the end of DNA fragments) though applying this method has currently some limits in terms of number of samples that can be sequenced parallelly.

Bibliographic reference :

Ellegard et al 2020 Communications biology <https://doi.org/10.1038/s42003-020-0899-z>

Minor comments:

- **3.3** L34: Change “resources” to “resource”

Done in the revised version

- **3.4** L38: Change “lakes” to “lake”

Done in the revised version

- **3.5** L38: What type of associated ecological functions?

We have added a few examples of important associated ecological functions: nutrient cycling, efficiency of trophic transfer, quality of fish production.

- **3.6** L42-44: Please add references.

We added a reference: Reid 2019 (Biological reviews) <https://doi.org/10.1111/brv.12480> ; this article documents 12 emerging threats to freshwater biodiversity that are either entirely new since 2006 or have since intensified (among which changing climates; harmful algal blooms; emerging contaminants; cumulative stressors etc); the effects of these stressors are evidenced for various biological groups, among which amphibians, fishes, invertebrates, waterbirds.

- **3.7** L65: Change “in the end” to “at the end”

Done in the revised version

- **3.8** L89: Change to “the robustness of the DNA results”

Done in the revised version

- **3.9** L93-95 and Fig. 1: I'm not sure what to make of these results. Are these differences between past and recent proxies for organic and biogenic compounds due to different conditions in the past/recent sediments or are they simply due to the effect of time on the past sediment? Are there models to suggest the sediments would have evolved that way with time or instead to counter the effect of time on these compounds?

We assume the differences between organic and biogenic compounds are mainly due to environmental changes between the past and present.

We cannot exclude that, when aging in sediments, a loss of pigments might have occurred. The preservation of pigments in lake sediments has previously been explored (Leavitt 1993; Verleyen et al. 2004 ; Buchaca and Catalan 2008). We know that the rate of pigment degradation in sediments is lower than in the water column (Leavitt 1993; Hurley and Armstrong 1990; Steenbergen et al. 1994; Villanueva and Hastings 2000). As reported by Leavitt (1993) the concentration of pigment in sediments is regulated by photo- and chemical oxidation, with 3 phases of loss: rapid oxidation in the water column ($T_{1/2}$ = days); slower post-depositional loss in surface sediments ($T_{1/2}$ = years); and very slow loss of double bonds in deeper sediments ($T_{1/2}$ = centuries). Post-depositional alterations occur on long timescales.

Moreover, among pigments, carotenoids (that we used in this study) are in general less labiles than chlorophylls (Leavitt 1993; Buchaca & Catalan 2008).

Several studies have demonstrated the value and relevance of these proxies (e.g. Taranu et al 2015, Engstrom et al 2005).

Moreover, given the time frame of our study (~150yrs) it is unlikely that these compounds underwent significant degradation.

Bibliographic references:

- Buchaca and Catalan 2008 *J Paleolimnol* (2008) 40:369–383 DOI 10.1007/s10933-007-9167-1
- Hurley & Armstrong 1990 *Limnol Oceanogr* 35:384–398 <https://doi.org/10.1139/f91-061>
- Leavitt 1993 *J Paleolimnol* 9:109–127 <https://doi.org/10.1007/BF00677513>
- Steenbergen et al 1994 *FEMS Microbiol Ecol* 13:335–351 <https://doi.org/10.1111/j.1574-6941.1994.tb00080.x>
- Taranu ZE et al. 2015 *Ecol. Lett.* 18, 375–384 doi:10.1111/ele.12420
- Verleyen et al. 2004 *Limnol. Oceanogr.*, 49(5), 1528-1539 <https://doi.org/10.4319/lo.2004.49.5.1528>
- Villanueva J, Hastings DW 2000 *Geochim Cosmochim Acta* 64:2281–2294

- **3.10** L105: Add “,” after “...Supplementary Fig. 1)”

Done in the revised version

- **3.11** L112-114: It would be nice to visualize these numbers, perhaps with a Venn diagram in the Supplementary Information.

We added a Venn diagram in the Supplementary Information as you suggested (Sup Fig.4).

- **3.12** L130-132: Please add the results of your PERMANOVA as a table in the Supplementary Information.

We added the PERMANOVA table in the SI.

- **3.13** L131: Change “OTUs” to “OTU”

Done in the revised version

- **3.14** L153: Change “associated to” to “associated with”

Done in the revised version

- **3.15** L153-154: “(i.e. a fine taxonomic level for which DNA modification during aging in sediments may be questioned)” <- This statement is unclear, please clarify.

The explanation regarding the relevance of OTUs delineation methods could be too long and not well appropriate in this section of the manuscript.

We therefore removed the parenthesis, and instead we added an explanation in the supplemental text. See the section ‘Methods S1’ p 5.

- **3.16** L161-162: Change “obligatory” to “obligate” and add “i.e. photosynthetics” to be consistent with the terminology used in Fig. 4.

Done in the revised version

- **3.17** L174: Change “which” to “whose”

Done in the revised version

- **3.18** L195: You are suggesting that microbial communities are inextinguishable. If that’s the case, would that mean that the communities found in the past sediments are also alive? How could we be sure that these represent a past community and not movement of live microbes through the sediment?

Maybe the sentence was not clear enough. We did not mean that all microbial taxa could live and survive everywhere. We meant that microbes are often seen as biological communities which are under no threat by anthropogenic disturbances or climate change because they are largely distributed in most of the environments. This is obviously a misconception to think that microbial communities are not affected by environmental stressors; we know that microbial groups are amongst the most relevant bioindicators to be used for the evaluation of water quality status and pollution. These biological groups are seen as inextinguishable but they are not, at least locally ; as pointed out by Bodelier et al 2011, microbial diversity is absent in the ongoing debates about global biodiversity loss and conservations policy, and it is crucial to open the microbial black box and progressively fill the knowledge gaps within the field of environmental microbiology. We have rephrased. Please see L 173-174 of the revised manuscript.

Regarding the risk of active eukaryotes in sediment, please see answer #3.2 & 3.1

Bibliographic reference:

-Bodelier et al 2011 *Front. Microbiol.*, 25 <https://doi.org/10.3389/fmicb.2011.00080>

● **3.19** L216-217: Is it really true that high altitude lakes are assumed to be less exposed to local anthropogenic pressures? I have doubts about this statement, please give a more recent reference to support the idea.

For historical reasons, human development is strongly correlated to topographic accessibility and elevation. We could find very few references which are lake-specific but we added two more recent and general references (Nogués-Bravo et al. 2008 and Leu et al. 2008). For instance Nogués-Bravo et al. (2008) show that human impact is larger in the lowlands and decreases almost monotonically with increased elevation in 13 of the largest mountain regions in the world (including the Alps and the Pyrenees, two mountain regions included in our study area).

Bibliographic references:

-Nogués-Bravo et al 2008 *Nature* 453, 216–219.

-Leu et al 2008 *Ecological Applications* 18, 1119–1139.

● **3.20** L217-218: This conclusion seems to be drawn a little quickly, especially considering that there is no data in this paper to support the fact that local human activities are the primary drivers of this temporal turnover (apart from the dating of the sediment, there is no measure of human activity in the lakes). If you are referring to TOC/pigment analysis to support this, please explain your reasoning and make it clearer. You could otherwise perhaps complement this statement with more data/analysis if you have any related to human activity.

Measuring the local impact of anthropogenic activities is complicated because human activities are highly diverse and can have complex ecological effects on lake ecosystems. Thus a general proxy that integrates the multiple dimensions of human activities is important and useful. Here, we used altitude as a covariable because altitude is recognized as a good proxy for human footprint on water quality in this region of the world (Müller et al. 1998, but see our response to #3.19). Moreover, altitude is an information available for all lakes which is not the case for other, perhaps more relevant, variables.

Nonetheless, we completely understand that the reader may want to see data supporting the statement that altitude is an appropriate proxy for human activities in our particular set of lakes. Following your comment, we decided to present additional data as supplementary information. As explained, monitoring data are scarce or simply inexistant for most of these lakes (especially for high-altitude lakes) and this prevents the development of a sophisticated human footprint index. Instead of that, we used remote sensing data to estimate the proportion of natural, artificial, and agricultural landscape in lake watersheds. Additionally we computed the human population living in a radius of 20 km around each lake. All these data support the idea that high altitude lakes (>1400m, i.e. the threshold identified by the regression tree) are much less subject to human-induced pressures than low- altitude lakes.

Please see Sup Figure 8

- **3.21** L243: How would you confirm that? Isn't there literature on this already?

Yet, the global drivers of freshwater C sequestration and its temporal variability have not been assessed at large scale. We can however mention some key publications, as for instance the recent article published in Science Advances by Anderson et al (2020). The authors reported the increases in lake C burial from the late 19th century, and showed that the majority (70%) of this increase is explained by rising N and P fertilizer use. The contemporary organic-C burial rates is also related to temperature control on burial rates, the increased aquatic production might be counteracted by temperature-dependent respiratory losses. However, the role of lake biological communities is not taken into account in these studies.

Independently from this literature focused on the global drivers of C sequestration, we know from numerous bibliographic references that food webs are important to understanding regulation of the C-cycle in lakes. Food web structure and nutrients can affect the carbon-emission/sequestration ratio and shift the state of the aquatic ecosystem between being a source or a sink for atmospheric carbon. One key bibliographic reference for lakes is the article published by Schindler et al 1997 in Science, demonstrating the influence of food web structure (regulated by top predators and nutrient loading) on carbon flux between lakes and the atmosphere.

Though we know that food web structure can determine many properties of aquatic ecosystems among which C fluxes and Production/Respiration balance (Schindler et al., 1997 ; Cole et al, 2000), as far as we know there is no long-term data demonstrating the link between biological community composition and C burial in lakes.

The message here was that there is a great potential with paleolimnological approaches (using multi-proxies) to explore the links between the dynamics of biodiversity and C burial and investigate the links with local and global pressures at centennial scales.

We added references on this topic: please see L234 - 236 of the revised manuscript.

References :

-Anderson et al 2020 Sci. Adv. 6 : eaaw2145 DOI: 10.1126/sciadv.aaw2145

-Cole et al 2000 Limnol. Oceanogr., 45(8), 2000, 1718–1730 <https://doi.org/10.4319/lo.2000.45.8.1718>

-Schindler et al 1997 Science 277, 248-251 DOI: 10.1126/science.277.5323.248

● **3.22** L288: Please be consistent throughout the manuscript and supplementary information and choose to use either “subsampling” or “sub-sampling”

We edited both the manuscript and supplementary information, and changed “sub-sampling” to “subsampling”.

● **3.23** L294-295: Please give a range for the depths at which the cores were collected.

The cores were collected at the deepest point in each lake which is reported in details in supplementary information Table 1. To make it clear we added the range and a reference to the SI in the text (Methods S2).

● **3.24** L.291-294: If the particle sizes of these elements (Pb, Cs) are comparable to microbial eukaryotes, could you use that as a calibration for leaching of microbes from one layer to another? Otherwise could pollen data help with calibration (if available)?

As previously mentioned (answer to comment #3.1 here before), gravitational percolation of water does not occur in lake sediments. Once sediments are compacted, vertical advection of pore water is minimal, and multivalent metals and organic compounds (pigments, polycyclic aromatic hydrocarbons, organic molecules with more than 15 carbon atoms) are immobilized in the sediment matrix (Anderson & Carpenter, 2011). Relevant temporal patterns have been observed in lake sediments (appropriate time-horizons) with no evidence of vertical leaching for organic molecules (Smol 2008; Meyers 2003).

We have no doubt on the fact that the strata we have analyzed here (top and bottom) represent deposits from two distant time periods. Several successful studies dealing with paleoreconstruction of microbial diversity in lakes (various types of lakes, various biological groups) have proved that those types of data, though not perfect as any other proxy, are relevant (focusing only on the last 3 years we could mention the publication by Konkel et al 2020; Stoof-Leichsenring et al. 2020 ; Belle & Parent 2019 ; Capo et al. 2019 ; Li et al. 2019 ; More et al. 2019 ; Pilon et al. 2019 ; Ruuskanen et al. 2019 ; Yan et al. 2019 ; Ahmed et al. 2018 ; Kisand et al. 2018 ; Marshall et al. 2018 ; Monchamp et al. 2018 ; More et al. 2018 ; Olajos et al. 2018 ; Tse et al. 2018 ; Capo et al. 2017 ; Dulias et al. 2017 ; Orsi et al. 2017 ; Vuillemin et al. 2017).

- **3.25** L379: Shouldn't DESeq2 be used on normalized data rather than raw count data (for example with the DESeq2 variance stabilizing transformation)?

Normalization is an integral part of the DESeq2 procedure. The starting point of a DESeq2 analysis as implemented by its authors is raw count of reads. From Love et al. (2014) *Beginner's guide to using the DESeq2 package*: "The count values must be raw counts of sequencing reads. This is important for DESeq2's statistical model to hold, as only the actual counts allow assessing the measurement precision correctly. Hence, please do not supply other quantities, such as (rounded) normalized counts, or counts of covered base pairs – this will only lead to nonsensical results."

- **3.26** Fig.1: I am not sure why the number of lakes isn't equal to 48 and varies among the different measurements taken (A, B and C). Please clarify.

In Fig 1 the number of lakes was not equal to 48 because some data were unfortunately not measured for a few lakes. We managed to complete the data for DNA, however there are 4 lakes missing for carotenoids and 2 missing for TOC. However, we believe the number of lakes is largely sufficient to provide good estimates of the changes between top and bottom. We clarified this in the Material & Methods section.

- **3.27** SI Fig.1: Is this from 48 samples extracted in duplicates? Please indicate average sequencing raw data read depth per sample.

We added the number of reads and OTUs per sample (or merged sample when relevant) to this figure.

- **3.28** SI Fig. 3 and Fig. 4: Please add stress values in legends for NMDS plots.

We added stress values for the two NMDS presented in the Supplementary Information.

AUTHORS message:

We would like to thank the reviewers for their thorough reviews. We appreciate the positive comments and the interest they have shown in our study. We also agree that there were several points in the manuscript that deserved further attention. We carefully considered each comment made by the reviewers and hereafter we answered and explained in detail how we revised the manuscript.

Important note: During the preparation of our response, we found out that two bottom samples (MAR and VER) were mislabeled. We fixed this problem in the new version and carefully double-checked the labeling of every sample. This change had only a marginal effect on the results. Importantly, none of the p-values (significance) and conclusions previously reported were affected. However, we believe it was ethically important to stress this point and we would like to apologize to the reviewers and the editor for this regrettable mistake.

Reviewers' Comments:

Reviewer #1:

Remarks to the Author:

The authors have carefully responded to all my comments. Below I noted some final linguistic suggestions:

Line 42, I would remove "but", and replace with "and" .. so "and see review..".

Line 43, I would put a comma after "However,"

Lines 51-52, I would add the "and specific pigments" after "chrysophytes" – so this order:

"groups able to produce morphological fossils (e.g. diatoms, chrysophytes) and specific pigments:

The way you have it now, it reads like only diatoms and chrysophytes have specific pigments, which is not true. But diatoms and chrysophytes have specific morphological fossils, which is what you are trying to say. So the slight change in order of words makes it clear.

Line 62, I would put a comma after "assemblages"

Line 67-68, Again minor but slight change in wording order would make it clearer. I would put "the Great Acceleration" earlier in sentence as the way it now reads, it seems like the Great Acceleration happened in 19th century. So, I would change it to:

Here, we used DNA records preserved in sediment to explore pre-impact (19th century) lake conditions chosen before the "Great Acceleration", a turning point in the Anthropocene from which the

Line 228, slight change in wording would make this clear: ("which" to "what" and add a "the" before "carbon"

required since we do not know what proportion of the carbon has a terrestrial origin in those lakes

Line 241, Euglenophyceae should be uppercase E

Line 271, occurrence is probably better word here than presence

Line 272, put a comma after "For instance," and a "the" before "Cercozoa"

John P. Smol

Reviewer #2:

Remarks to the Author:

In all honesty, I think the readability of the paper has slightly decreased and the new additions caused grammatical issues that need to be fixed first before the paper can be published. These are all minor things though, but I am surprised that it happened at this stage:

Line 42: remove "but" before "see review by Smol"

Line 43. Place comma after However.

Line 72. "representing reference conditions" has been placed in the wrong spot in that sentence.

Line 83. Do not start the sentence with "Finally".

Line 88: Others have shown and stated this already since early 2000's. We know that sed aDNA complements traditional proxies from at least 25 other studies now.

Line 93: please complete the sentence by saying what is being compared.

Line 100: I do not follow the new sentence "with usual reserves due to..."

Line 146: change to with an increase of obligate photoautotrophs.

Line 147: On the other hand. Replace by In addition

Line 174: are you talking about microbial prokaryotes or eukaryotes?

Line 224: Obligate photosynthetics is not the correct term: Use Obligate oxygenic phototautotrophs instead.

Line 226 with new addition does not read well. Please rephrase and split into two sentences. I would start the sentence in 225 with "Our combined DNA and pigment-based results..." and forget about the interpretation of the TOC content because that is indeed irrelevant unless you generated a detailed biomarker survey and BIT index to see how much of the TOC is terrestrial vs. water column derived. Just delete the new red text.

Line 236: please check the grammar here. Your new additions are sometimes a bit sloppy. Again, since you did not perform a detailed biomarker survey with a BIT index you cannot determine the origin of this organic matter and say what caused the increase in TOC content.

Line 239-241: hasty developed sentence, incorrect grammar.

Line 250: This is also no longer a working sentence with the comma and the

semicolon.

Line 270-276: The readability of this new addition is poor. Please rewrite using proper English grammar! I also feel that in general the readability has gone down as the additions now appear to fall from the sky a bit in an attempt to please the reviewers.

Reviewer #3:

Remarks to the Author:

The authors did a great job addressing the three reviewers' comments thoroughly and adding a lot of important information about their methods. My main concern was about the use of sedDNA for microbes specifically because, as the authors themselves say in their answer to comment 3.2, prokaryotes can be found living in deep sediment (as opposed to other taxa that would not survive in such conditions). Although I understand this is less likely to be the case for micro-eukaryotes, it made me question whether we could infer whether organisms found in deeper sediment were truly ancient. While they have convinced me that leaching of microbial DNA through lake sediment is unlikely, they have not made it quite clear that this is specifically related to microbes in the additional text provided in the supplementary file.

I do not mean for the authors to make any major modifications to the paper but would appreciate it if they could use microbial references in their justification of the use of sedDNA in the paragraph they added about this in section Methods S1 page 4 — perhaps Ellegard et al 2020 (I found this quite relevant: "Naturally, most bacterial sed-eDNA studies have focussed on phototrophs that are allochthonous to the deep sediment, including cyanobacteria, Chlorobi and Chromatiaceae") or other papers referenced in their answer to comment 3.24? These would seem more relevant than the references added (8-10), which, while supporting the use of sedDNA in lake sediments, are not related to microbial communities.

I also found that the authors addressed my concerns very well in their responses but less so in the text they added. I think they could provide some of the explanations they gave me in the supplementary (or main) text, for example with the ideas conveyed in these two answers:

3.1 — "We cannot completely exclude the presence of active cells in compacted sediments; however, if we exclude resting cells considered as dormant but revivifiable (Ellegard et al 2020), and the taxa living at the surface of sediment

(benthic protists in oxygenated zones), then, the rest of potentially active eukaryotic cells are most probably rare extremophile specialists representing a negligible fraction of the immense diversity of the total micro-eukaryote community. None of the major groups we identified to explain the changes between Top and Bottom samples are known to live in deep sediments.”

3.2 — “The micro-eukaryotic taxa we detected are mainly known as planktonic groups ; some (heterotrophs) that can live at the surface of oxygenated sediment (as reported in the revised manuscript; e.g L270-276) were also found, but these taxa do not survive in hypoxic/anoxic conditions.”

- **REVIEWERS' COMMENTS: POINT BY POINT ANSWERS**

Reviewer #1 (Remarks to the Author):

The authors have carefully responded to all my comments. Below I noted some final linguistic suggestions:

Line 42, I would remove “but”, and replace with “and” .. so “and see review..”.

=>modified as suggested

Line 43, I would pit a comma after “However,”

=>modified as suggested

Lines 51-52, I would add the “and specific pigments” after “chrysophytes0” – so this order: “groups able to produce morphological fossils (e.g. diatoms, chrysophytes) and specific pigments: The way you have it now, it reads like only diatoms and chrysophytes have specific pigments, which is not true. But diatoms and chrysophytes have specific morphological fossils, which is what you are trying to say. So the slight change in order of words makes it clear.

=>modified as suggested

Line 62, I would put a comma after “assemblages”

=>modified as suggested

Line 67-68, Again minor but slight change in wording order would make it clearer. I would put “the Great Acceleration” earlier in sentence as the way it now reads, it seems like the Great Acceleration happened in 19th century. So, I would change it to:

Here, we used DNA records preserved in sediment to explore pre-impact (19th century) lake conditions chosen before the “Great Acceleration”, a turning point in the Anthropocene from

=>modified as suggested

Line 228, slight change in wording would make this clear: (“which” to “what” and add a “the” before “carbon” Required since we do not know what proportion of the carbon has a terrestrial origin in those lakes

=>modified as suggested

Line 241, Euglenophyceae should be uppercase E

=>modified as suggested

Line 271, occurrence is probably better word here than presence

=>modified as suggested

Line 272, put a comma after “For instance,” and a “the” before “Cercozoa”

=>modified as suggested

John P. Smol

Reviewer #2 (Remarks to the Author):

In all honesty, I think the readability of the paper has slightly decreased and the new additions caused grammatical issues that need to be fixed first before the paper can be published. These are all minor things though, but I am surprised that it happened at this stage:

Line 42: remove “but” before “see review by Smol”
=>modified as suggested

Line 43. Place comma after However.
=>modified as suggested

Line 72. “representing reference conditions” has been placed in the wrong spot in that sentence.
=>This was modified as suggested
Modified as follows : whereby DNA preserved in recent sediment deposits (i.e. top) representing modern biological assemblages is compared to a sediment sample representing reference conditions from the 19th century (i.e. bottom).

Line 83. Do not start the sentence with “Finally”.
=>modified as suggested

Line 88: Others have shown and stated this already since early 2000’s. We know that sed aDNA complements traditional proxies from at least 25 other studies now.
=> the sentence was modified accordingly
Modified as follows : DNA preserved in lake sediment is an essential tool

Line 93: please complete the sentence by saying what is being compared.
=> the sentence was modified, the comparison done between two periods (recent versus past) is clear in other sections ; here, in the methods’ description, we only need to describe how we proceeded to perform the subsampling of sediment.
Initial sentence : two sediment layers were subsampled and analysed to perform the comparison between recent and past times
Modified as follows : two sediment layers, i.e. a recent sediment deposit and a sediment deposit from the 19th century, were subsampled and preserved for downstream analyses.

Line 100: I do not follow the new sentence “with usual reserves due to...”
=> the sentence was modified.
Initial sentence : ..., suggesting a lower biological production in the past, unless (with usual reserves due to the potential loss of a fraction of organic material affected by diagenetic processes).
Modified as follows : This suggests a lower biological production in the past, as long as the diagenetic processes occurring during burying in sediments had a limited effect on the loss of organic material.

Line 146: change to with an increase of obligate photoautotrophs.
=>modified as suggested

Line 147: On the other hand. Replace by In addition
=>modified as suggested

Line 174: are you talking about microbial prokaryotes or eukaryotes?
=> We talk about all types of microbes (prokaryotes and eukaryotes) it’s why we have not specified, and have used a general term ‘microbes’

Line 224: Obligate photosynthetics is not the correct term: Use Obligate oxygenic phototototrophs instead.

=>modified as suggested

Line 226 with new addition does not read well. Please rephrase and split into two sentences. I would start the sentence in 225 with "Our combined DNA and pigment-based results...." and forget about the interpretation of the TOC content because that is indeed irrelevant unless you generated a detailed biomarker survey and BIT index to see how much of the TOC is terrestrial vs. water column derived. Just delete the new red text.

=> the sentence was modified as suggested.

Modified as follows : Our combined DNA and pigment-based results obtained for 48 lakes are in line with several paleolimnological studies that have reported an increase in primary production throughout the Holocene

Line 236: please check the grammar here. Your new additions are sometimes a bit sloppy. Again, since you did not perform a detailed biomarker survey with a BIT index you cannot determine the origin of this organic matter and say what caused the increase in TOC content.

=> the sentence was modified as suggested.

Initial sentence : If confirmed, this would mean that anthropogenically-driven changes in these assemblages which are critical to biogeochemical cycles might have affected the carbon cycle in lakes, as for instance the increase of lake C burial observed from the late 19th century.

Modified as follows : If confirmed, this would mean that anthropogenically-driven changes in these biological assemblages might have affected the carbon cycle in lakes, with a potential contribution to the increase of lake C burial observed from the late 19th century.

Line 239-241: hasty developed sentence, incorrect grammar.

=> the sentence was modified.

Initial sentence : Known trophic status indices use Chlorophyceae as a typical algal groups for eutrophic conditions (with Cyanobacteria, and Euglenophyceae, e.g.

Modified as follows : Standardized phytoplankton-based indices used to evaluate lake trophic status generally use Chlorophyceae as a typical algal groups for eutrophic conditions

Line 250: This is also no longer a working sentence with the comma and the semicolon.

=> the sentence was modified.

Initial sentence : Mixotrophy is a common trophic strategy among unicellular eukaryotes, the harsh growth conditions found in high mountain lakes seem in particular to favor mixotrophs; however there is currently little evidence for how mixotrophy varies across environmental gradients.

Modified as follows : Mixotrophy is a common trophic strategy among unicellular eukaryotes. Although the harsh growth conditions found in high mountain lakes seem in particular to favor mixotrophs⁵⁰, there is currently little evidence for how mixotrophy varies across environmental gradients⁵¹.

Line 270-276: The readability of this new addition is poor. Please rewrite using proper English grammar! I also feel that in general the readability has gone down as the additions now appear to fall from the sky a bit in an attempt to please the reviewers.

=> This section was rephrased to improve clarity

Initial sentence : A number of gaps exist in our understanding of microbial diversity patterns and their long-term dynamics. This is true in particular for heterotrophic protists. Here we showed the possibility to track the past occurrence and dynamics of various heterotrophic groups which are constituents of

both pelagic and benthic food webs. For instance, changes in the Cercozoa community were detected, with a decrease in recent periods of taxa living on oxygenated surface sediments, which tends to confirm the modification of benthic food webs affected by the general expansion of hypoxia in lakes during the last centuries⁶¹ with impacts on lake functioning and recycling of organic matter and nutrients⁴².

Modified as follows : A number of gaps exist in our understanding of microbial diversity patterns and their long-term dynamics. This is true in particular for heterotrophic protists which are often missing in lake biomonitoring and aquatic biogeochemical models. Here we showed the possibility to track the past occurrence and dynamics of various heterotrophic groups which are constituents of both pelagic and benthic food webs. This allows providing unique information in terms of lake functional ecology. For instance, our data suggest a decrease of taxa living on oxygenated surface sediments (e.g. Cercozoa) in recent periods, which tends to confirm that benthic food webs were affected by the general expansion of hypoxia in lakes during the last centuries⁶¹.

Reviewer #3 (Remarks to the Author):

The authors did a great job addressing the three reviewers' comments thoroughly and adding a lot of important information about their methods. My main concern was about the use of sedDNA for microbes specifically because, as the authors themselves say in their answer to comment 3.2, prokaryotes can be found living in deep sediment (as opposed to other taxa that would not survive in such conditions). Although I understand this is less likely to be the case for micro-eukaryotes, it made me question whether we could infer whether organisms found in deeper sediment were truly ancient. While they have convinced me that leaching of microbial DNA through lake sediment is unlikely, they have not made it quite clear that this is specifically related to microbes in the additional text provided in the supplementary file.

I do not mean for the authors to make any major modifications to the paper but would appreciate it if they could use microbial references in their justification of the use of sedDNA in the paragraph they added about this in section Methods S1 page 4 — perhaps Ellegard et al 2020 (I found this quite relevant: “Naturally, most bacterial sed-eDNA studies have focussed on phototrophs that are allochthonous to the deep sediment, including cyanobacteria, Chlorobi and Chromatiaceae”) or other papers referenced in their answer to comment 3.24? These would seem more relevant than the references added (8-10), which, while supporting the use of sedDNA in lake sediments, are not related to microbial communities.

=> this is now included in the Methods S1 (see below)

I also found that the authors addressed my concerns very well in their responses but less so in the text they added. I think they could provide some of the explanations they gave me in the supplementary (or main) text, for example with the ideas conveyed in these two answers:

3.1 — “We cannot completely exclude the presence of active cells in compacted sediments; however, if we exclude resting cells considered as dormant but revivable (Ellegard et al 2020), and the taxa living at the surface of sediment (benthic protists in oxygenated zones), then, the rest of potentially active eukaryotic cells are most probably rare extremophile specialists representing a negligible fraction of the immense diversity of the total micro-eukaryote community. None of the major groups we identified to explain the changes between Top and Bottom samples are known to live in deep sediments.”

3.2 — “The micro-eukaryotic taxa we detected are mainly known as planktonic groups ; some (heterotrophs) that can live at the surface of oxygenated sediment (as reported in the revised manuscript; e.g L270-276) were also found, but these taxa do not survive in hypoxic/anoxic conditions.”

=> We have now included these information in the Methods S1

« The presence of active cells in sediments cannot be excluded; however, if we exclude resting cells considered as dormant but revivifiable (Ellegard et al 2020), and the taxa living at the surface of sediment (benthic protists in oxygenated zones), then, the rest of potentially active eukaryotic cells are most probably rare extremophile specialists representing a negligible fraction of the immense diversity of the total micro-eukaryote community. The micro-eukaryotic taxa we detected in this study are mainly known as planktonic groups ; some (heterotrophs) that can live at the surface of oxygenated sediment were also found, but these taxa do not survive in hypoxic/anoxic conditions. From our data we could not identify any taxa known to live in deep sediments »